# Contrastive Behavioral Similarity Embeddings for Generalization in Reinforcement Learning

**Rishabh Agarwal**[*]  **Marlos C. Machado**[‡]  **Pablo Samuel Castro**  **Marc G. Bellemare**
Google Research, Brain Team
`{rishabhagarwal, marlosm, psc, bellemare}@google.com`

## Abstract

Reinforcement learning methods trained on few environments rarely learn policies that generalize to unseen environments. To improve generalization, we incorporate the inherent sequential structure in reinforcement learning into the representation learning process. This approach is orthogonal to recent approaches, which rarely exploit this structure explicitly. Specifically, we introduce a theoretically motivated *policy similarity metric* (PSM) for measuring behavioral similarity between states. PSM assigns high similarity to states for which the optimal policies in those states as well as in future states are similar. We also present a contrastive representation learning procedure to embed any state similarity metric, which we instantiate with PSM to obtain *policy similarity embeddings* (PSEs[1]). We demonstrate that PSEs improve generalization on diverse benchmarks, including LQR with spurious correlations, a jumping task from pixels, and Distracting DM Control Suite.

## 1 Introduction

Current reinforcement learning (RL) approaches often learn policies that do not generalize to environments different than those the agent was trained on, even when these environments are semantically equivalent (Tachet des Combes et al., 2018; Song et al., 2019; Cobbe et al., 2019). For example, consider a jumping task where an agent, learning from pixels, needs to jump over an obstacle (Figure 1). Deep RL agents trained on a few of these tasks with different obstacle positions struggle to solve test tasks where obstacles are at previously unseen locations.

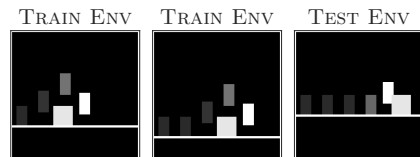

Figure 1: **Jumping task**: The agent (white block), learning from pixels, needs to jump over an obstacle (grey square). The challenge is to generalize to unseen obstacle positions and floor heights in test tasks using a small number of training tasks. We show the agent's trajectories using faded blocks.

Recent solutions to circumvent poor generalization in RL are adapted from supervised learning, and, as such, largely ignore the sequential aspect of RL. Most of these solutions revolve around enhancing the learning process, including data augmentation (*e.g.,* Kostrikov et al., 2020; Lee et al., 2020a), regularization (Cobbe et al., 2019; Farebrother et al., 2018), noise injection (Igl et al., 2019), and diverse training conditions (Tobin et al., 2017); they rarely exploit properties of the sequential decision making problem such as similarity in actions across temporal observations.

Instead, we tackle generalization by incorporating properties of the RL problem into the representation learning process. Our approach exploits the fact that an agent, when operating in environments with similar underlying mechanics, exhibits at least short sequences of behaviors that are similar across these environments. Concretely, the agent is optimized to learn an embedding in which states are close when the agent's optimal policies in these states and future states are similar. This notion of proximity is general and it is applicable to observations from different environments.

Specifically, inspired by bisimulation metrics (Castro, 2020; Ferns et al., 2004), we propose a novel *policy similarity metric* (PSM). PSM (Section 3) defines a notion of similarity between states originated from different environments by the proximity of the long-term optimal behavior from these states. PSM is reward-agnostic, making it more robust for generalization compared to approaches that

---

[*]Also at Mila, Université de Montréal. [‡]Now at DeepMind.
[1]Pronounce 'pisces'.

rely on reward information. We prove that PSM yields an upper bound on suboptimality of policies transferred from one environment to another (Theorem 1), which is not attainable with bisimulation.

We employ PSM for representation learning and introduce *policy similarity embeddings* (PSEs) for deep RL. To do so, we present a general contrastive procedure (Section 4) to learn an embedding based on any state similarity metric. PSEs are the instantiation of this procedure with PSM. PSEs are appealing for generalization as they encode task-relevant invariances by putting behaviorally equivalent states together. This is unlike prior approaches, which rely on capturing such invariances without being *explicitly* trained to do so, for example, through value function similarities across states (*e.g.,* Castro & Precup, 2010), or being robust to fixed transformations of the observation space (*e.g.,* Kostrikov et al., 2020; Laskin et al., 2020a).

PSEs lead to better generalization while being orthogonal to how most of the field has been tackling generalization. We illustrate the efficacy and broad applicability of our approach on three existing benchmarks specifically designed to test generalization: (i) jumping task from pixels (Tachet des Combes et al., 2018) (Section 5), (ii) LQR with spurious correlations (Song et al., 2019) (Section 6.1), and (iii) Distracting DM Control Suite (Stone et al., 2021; Zhang et al., 2018b) (Section 6.2). Our approach improves generalization compared to a wide variety of approaches including standard regularization (Farebrother et al., 2018; Cobbe et al., 2019), bisimulation (Castro & Precup, 2010; Castro, 2020; Zhang et al., 2021), out-of-distribution generalization (Arjovsky et al., 2019) and state-of-the-art data augmentation (Kostrikov et al., 2020; Lee et al., 2020a).

## 2 PRELIMINARIES

We describe an environment as a Markov decision process (MDP) (Puterman, 1994) $\mathcal{M} = (\mathcal{X}, \mathcal{A}, R, P, \gamma)$, with a state space $\mathcal{X}$, an action space $\mathcal{A}$, a reward function $R$, transition dynamics $P$, and a discount factor $\gamma \in [0, 1)$. A policy $\pi(\cdot \mid x)$ maps states $x \in \mathcal{X}$ to distributions over actions. Whenever convenient, we abuse notation and write $\pi(x)$ to describe the probability distribution $\pi(\cdot \mid x)$, treating $\pi(x)$ as a vector. In RL, the goal is to find an optimal policy $\pi^*$ that maximizes the cumulative expected return $\mathbb{E}_{a_t \sim \pi(\cdot \mid x_t)}[\sum_t \gamma^t R(x_t, a_t)]$ starting from an initial state $x_0$.

We are interested in learning a policy that generalizes across related environments. We formalize this by considering a collection $\rho$ of MDPs, sharing an action space $\mathcal{A}$ but with disjoint state spaces. We use $\mathcal{X}$ and $\mathcal{Y}$ to denote the state spaces of specific environments, and write $R_{\mathcal{X}}$, $P_{\mathcal{X}}$ for the reward and transition functions of the MDP whose state space is $\mathcal{X}$, and $\pi^*_{\mathcal{X}}$ for its optimal policy, which we assume unique without loss of generality. For a given policy $\pi$, we further specialize these into $R^\pi_{\mathcal{X}}$ and $P^\pi_{\mathcal{X}}$, the reward and state-to-state transition dynamics arising from following $\pi$ in that MDP.

We write $\mathcal{S}$ for the union of the state spaces of the MDPs in $\rho$. Concretely, different MDPs correspond to specific scenarios in a problem class (Figure 1), and $\mathcal{S}$ is the space of all possible configurations. Used without subscripts, $R$, $P$, and $\pi$ refer to the reward and transition function of this "union MDP", and a policy defined over $\mathcal{S}$; this notation simplifies the exposition. We measure distances between states across environments using pseudometrics[2] on $\mathcal{S}$; the set of all such pseudometrics is $\mathbb{M}$, and $\mathbb{M}_p$ is the set of metrics on probability distributions over $\mathcal{S}$.

In our setting, the learner has access to a collection of training MDPs $\{\mathcal{M}_i\}_{i=1}^N$, drawn from $\rho$. After interacting with these environments, the learner must produce a policy $\pi$ over the entire state space $\mathcal{S}$, which is then evaluated on unseen MDPs from $\rho$. Similar in spirit to the setting of transfer learning (Taylor & Stone, 2009), here we evaluate the policy's zero-shot performance on $\rho$.

Our policy similarity metric (Section 3) builds on the concept of **$\pi$-bisimulation** (Castro, 2020). Under the $\pi$-bisimulation metric, the distance between two states, $x$ and $y$, is defined in terms of the difference between the expected rewards obtained when following policy $\pi$. The $\pi$-bisimulation metric $d_\pi$ satisfies a recursive equation based on the 1-Wasserstein metric $\mathcal{W}_1 : \mathbb{M} \to \mathbb{M}_p$, where $\mathcal{W}_1(d)(A, B)$ is the minimal cost of transporting probability mass from $A$ to $B$ (two probability distributions on $\mathcal{S}$) under the base metric $d$ (Villani, 2008). The recursion is

$$d_\pi(x, y) = |R^\pi(x) - R^\pi(y)| + \gamma \mathcal{W}_1(d_\pi)\big(P^\pi(\cdot \mid x), P^\pi(\cdot \mid y)\big), \qquad x, y \in \mathcal{S}. \tag{1}$$

To achieve good generalization properties, we learn an embedding function $z_\theta : \mathcal{S} \to \mathbb{R}^k$ that reflects the information encoded in the policy similarity metric; this yields a policy similarity

---

[2]Pseudometrics are generalization of metrics where the distance between two distinct states can be zero.

embedding (Section 4). We use contrastive methods (Hadsell et al., 2006; Oord et al., 2018) whose track record for representation learning is well-established. We adapt **SimCLR** (Chen et al., 2020), a popular contrastive method for learning embeddings of image inputs. Given two inputs $x$ and $y$, their *embedding similarity* is $s_\theta(x, y) = sim(z_\theta(x), z_\theta(y))$, where $sim(u, v) = \frac{u^T v}{\|u\|\|v\|}$ denotes the cosine similarity function. SimCLR aims to maximize similarity between augmented versions of an image (*e.g.,* crops, colour changes) while minimizing its similarity to other images. The loss used by SimCLR for two versions $x, y$ of an image, and a set $\mathcal{X}'$ containing other images is:

$$\ell_\theta(x, y; \mathcal{X}') = -\log \frac{\exp(\lambda s_\theta(x, y))}{\exp(\lambda s_\theta(x, y)) + \sum_{x' \in \mathcal{X}' \setminus \{x\}} \exp(\lambda s_\theta(x', y))} \qquad (2)$$

where $\lambda$ is an inverse temperature hyperparameter. The overall SimCLR loss is then the expected value of $\ell_\theta(x, y; S)$, when $x$, $y$, and $\mathcal{X}'$ are drawn from some augmented training distribution.

## 3   POLICY SIMILARITY METRIC

A useful tool in learning a policy that generalizes is to understand which states result in similar behavior, and which do not. To be maximally effective, this similarity should go beyond the immediately chosen action and consider long-term behavior. In this regards, the $\pi$-bisimulation metrics are interesting as they are based on the full sequence of future rewards received from different states. However, considering rewards can be both too restrictive (when the policies are the same, but the obtained rewards are not; see Figure D.1) or too permissive (when the policies are different, but the obtained rewards are not; see Figure 3a). In fact, $\pi$-bisimulation metrics actually lead to poor generalization in our experiments (Sections 5.1 and 5.2).

To address this issue, we instead consider the similarity between policies themselves. We replace the absolute reward difference by a probability pseudometric between policies, denoted DIST. Additionally, since we would like to perform well in unseen environments, we are interested in similarity in *optimal* behavior. We thus use $\pi^*$ as the grounding policy. This yields the *policy similarity metric* (PSM), for which states are close when the optimal policies in these states and future states are similar. For a given DIST, the PSM $d^* : \mathcal{S} \times \mathcal{S} \to \mathbb{R}$ satisfies the recursive equation

$$d^*(x, y) = \underbrace{\text{DIST}\big(\pi^*(x), \pi^*(y)\big)}_{(A)} + \underbrace{\gamma \mathcal{W}_1(d^*)\big(P^{\pi^*}(\cdot \,|\, x), P^{\pi^*}(\cdot \,|\, y)\big)}_{(B)}. \qquad (3)$$

The DIST term captures the difference in local optimal behavior (A) while $\mathcal{W}_1$ captures long-term optimal behavior difference (B); the exact weights assigned to the two are given by the discount factor. Furthermore, when DIST is bounded, $d^*$ is guaranteed to be finite. While there are technically multiple PSMs (one for each DIST), we omit this distinction whenever clear from context. A proof of the uniqueness of $d^*$ is given in Proposition B.1.

Our main use of PSM will be to compare states across environments. In this context, we identify the terms in Equation 3 with specific environments for clarity and write (despite its technical inaccuracy)

$$d^*(x, y) = \text{DIST}\big(\pi_{\mathcal{X}}^*(x), \pi_{\mathcal{Y}}^*(y)\big) + \gamma \mathcal{W}_1\big(d^*\big)(P_{\mathcal{X}}^{\pi^*}(\cdot \,|\, x), P_{\mathcal{Y}}^{\pi^*}(\cdot \,|\, y)).$$

PSM is applicable to both discrete and continuous action spaces. In our experiments, DIST is the total variation distance ($TV$) when $\mathcal{A}$ is discrete, and we use the $\ell_1$ distance between the mean actions of the two policies when $\mathcal{A}$ is continuous. PSM can be computed iteratively using dynamic programming (Ferns et al., 2011) (more details in Section D.1). Furthermore, when $\pi^*$ is unavailable on training environments, we replace it by an approximation $\hat{\pi}^*$ to obtain an approximate PSM, which is close to the exact PSM depending on the suboptimality of $\hat{\pi}^*$ (Proposition D.3).

Despite resembling $\pi$-bisimulation metrics in form, the PSM possesses different characteristics which are better suited to the problem of generalizing a learned policy. To illustrate this point, consider the following simple nearest-neighbour scheme: Given a state $y \in \mathcal{Y}$, denote its closest match in $\mathcal{X}$ by $\tilde{x}_y := \arg\min_{x \in \mathcal{X}} d^*(x, y)$. Suppose that we use this scheme to transfer $\pi_{\mathcal{X}}^*$ to $\mathcal{M}_{\mathcal{Y}}$, in the sense that we behave according to the policy $\tilde{\pi}(y) = \pi_{\mathcal{X}}^*(\tilde{x}_y)$. We can then bound the difference between $\tilde{\pi}$ and $\pi_{\mathcal{Y}}^*$, something that is not possible if $d^*$ is replaced by a $\pi$-bisimulation metric.

**Theorem 1.** *[Bound on policy transfer] For any $y \in \mathcal{Y}$, let $Y_y^t \sim P^{\tilde{\pi}}(\cdot \,|\, Y_y^{t-1})$ define the sequence of random states encountered starting in $Y_y^0 = y$ and following policy $\tilde{\pi}$. We have:*

$$\mathbb{E}_{Y_y^t}\left[\sum_{t\geq 0}\gamma^t TV\left(\tilde{\pi}(Y_y^t),\pi^*(Y_y^t)\right)\right]\leq \frac{1+\gamma}{1-\gamma}d^*(\tilde{x}_y,y)\,.$$

The proof is in the Appendix (Section B). Theorem 1 is non-vacuous whenever $d^*(\tilde{x}_y,y)<1/(1+\gamma)$. In particular, $d^*(\tilde{x}_y,y)=0$ implies that the transferred policy is optimal. Although this scheme is not practical (computing $d^*$ requires knowledge of $\pi_{\mathcal{Y}}^*$), it shows that meaningful policy generalization can be obtained if we can find a mapping that generalizes across $\mathcal{S}$. Put another way, PSM gives us a principled way of lifting generalization across inputs (a supervised learning problem) to generalization across environments. We now describe how to employ PSM for learning representations that put together states in which the agent's long-term optimal behavior is similar.

## 4 LEARNING CONTRASTIVE METRIC EMBEDDINGS

To generalize a learned policy to new environments, we build on the success of contrastive representations (Section 2). Given a state similarity metric $d$, we develop a general procedure (Algorithm 1) to learn contrastive metric embeddings (CMEs) for $d$. We utilize the metric $d$ for defining the set of positive and negative pairs, as well as assigning importance weights to these pairs in the contrastive loss (Equation 4).

---
**Algorithm 1** Contrastive Metric Embeddings (CMEs)

1: **Given**: State embedding $z_\theta(\cdot)$, Metric $d(\cdot,\cdot)$ Training environments $\{\mathcal{M}_i\}_{i=1}^N$. **Hyperparameters**: Temperature $1/\lambda$, Scale $\beta$, Total training steps $K$
2: **for** each step $k=1,\ldots,K$ **do**
3:    Sample a pair of training MDPs $\mathcal{M}_\mathcal{X},\mathcal{M}_\mathcal{Y}$
4:    Update $\theta$ to minimize $\mathscr{L}_{\text{CME}}$ where
     $\mathscr{L}_{\text{CME}}=\mathbb{E}_{\mathcal{M}_\mathcal{X},\mathcal{M}_\mathcal{Y}\sim\rho}[L_\theta(\mathcal{M}_\mathcal{X},\mathcal{M}_\mathcal{Y})]$
5: **end for**

---

We first apply a transformation to convert $d$ to a similarity measure $\Gamma$, bounded in [0, 1] for "soft" similarities. In this work, we transform $d$ using the Gaussian kernel with a positive scale parameter $\beta$, that is, $\Gamma(x,y)=\exp(-d(x,y)/\beta)$. $\beta$ controls the sensitivity of the similarity measure to $d$.

Second, we select the positive and negative pairs given a set of states $\mathcal{X}'\subseteq\mathcal{X}$ and $\mathcal{Y}$ from MDPs $\mathcal{M}_\mathcal{X},\mathcal{M}_\mathcal{Y}$ respectively. For each anchor state $y\in\mathcal{Y}$, we use its nearest neighbor in $\mathcal{X}'$ based on the similarity measure $\Gamma$ to define the positive pairs $\{(\tilde{x}_y,y)\}$, where $\tilde{x}_y=\underset{x\in\mathcal{X}'}{\arg\max}\,\Gamma(x,y)$. The remaining states in $\mathcal{X}'$, paired with $y$, are used as negative pairs. This choice of pairs is motivated by Theorem 1, which shows that if we transfer the optimal policy in $\mathcal{M}_\mathcal{X}$ to the nearest neighbors defined using PSM, its performance in $\mathcal{M}_\mathcal{Y}$ has suboptimality bounded by PSM.

Next, we define a soft version of the SimCLR contrastive loss (Equation 2) for learning the function $z_\theta$, which maps states (usually high-dimensional) to embeddings. Given a positive state pair $(\tilde{x}_y,y)$, the set $\mathcal{X}'$, and the similarity measure $\Gamma$, the loss (pseudocode provided in Section J.2) is given by

$$\ell_\theta(\tilde{x}_y,y;\mathcal{X}')=-\log\frac{\Gamma(\tilde{x}_y,y)\exp(\lambda s_\theta(\tilde{x}_y,y))}{\Gamma(\tilde{x}_y,y)\exp(\lambda s_\theta(\tilde{x}_y,y))+\sum_{x'\in\mathcal{X}'\setminus\{\tilde{x}_y\}}(1-\Gamma(x',y))\exp(\lambda s_\theta(x',y))}\quad(4),$$

where we use the same notation as in Equation 2. Following SimCLR, we use a non-linear projection of the representation as $z_\theta$ (Figure A.1). The agent's policy is an affine function of the representation.

The total contrastive loss for $\mathcal{M}_\mathcal{X}$ and $\mathcal{M}_\mathcal{Y}$ utilizes the optimal trajectories $\tau_\mathcal{X}^*=\{x_t\}_{t=1}^N$ and $\tau_\mathcal{Y}^*=\{y_t\}_{t=1}^N$, where $x_{t+1}\sim P_\mathcal{X}^{\pi^*}(\cdot\,|\,x_t)$ and $y_{t+1}\sim P_\mathcal{Y}^{\pi^*}(\cdot\,|\,y_t)$. We set $\mathcal{X}'=\tau_\mathcal{X}^*$ and define

$$L_\theta(\mathcal{M}_\mathcal{X},\mathcal{M}_\mathcal{Y})=\mathbb{E}_{y\sim\tau_\mathcal{Y}^*}[\ell_\theta(\tilde{x}_y,y;\tau_\mathcal{X}^*)]\qquad\text{where }\tilde{x}_y=\underset{x\in\tau_\mathcal{X}^*}{\arg\max}\,\Gamma(x,y).$$

We refer to CMEs learned with policy similarity metric as *policy similarity embeddings (PSEs)*. PSEs can trivially be combined with data augmentation by using augmented states when computing state embeddings. We simultaneously learn PSEs with the RL agent by adding $\mathscr{L}_{\text{CME}}$ (Algorithm 1) as an auxiliary objective during training. Next, we illustrate the benefits from this auxiliary objective.

## 5 JUMPING TASK FROM PIXELS: A CASE STUDY

**Task Description**. The jumping task (Tachet des Combes et al., 2018) (Figure 1) captures, using well-defined factors of variations, whether agents can learn the correct invariances required for generalization, directly from image inputs. The task consists of an agent trying to jump over an obstacle. The agent has access to two actions: *right* and *jump*. The agent needs to time the jump

Table 1: Percentage (%) of test tasks solved by different methods without and with data augmentation. The "wide", "narrow", and random grids are described in Figure 2. We report average performance across 100 runs with different random initializations, with standard deviation between parentheses.

| Data Augmentation | Method | Grid Configuration (%) | | |
|---|---|---|---|---|
| | | "Wide" | "Narrow" | Random |
| ✗ | Dropout and $\ell_2$ reg. | 17.8 (2.2) | 10.2 (4.6) | 9.3 (5.4) |
| | Bisimulation Transfer[4] | 17.9 (0.0) | **17.9** (0.0) | 30.9 (4.2) |
| | PSEs | **33.6** (10.0) | 9.3 (5.3) | **37.7** (10.4) |
| ✓ | RandConv | 50.7 (24.2) | 33.7 (11.8) | 71.3 (15.6) |
| | RandConv + Bisimulation | 41.4 (17.6) | 17.4 (6.7) | 33.4 (15.6) |
| | RandConv + PSEs | **87.0** (10.1) | **52.4** (5.8) | **83.4** (10.1) |

| (a) "Wide" grid | (b) "Narrow" grid | (c) Random grid |

Figure 2: **Jumping Task**: Visualization of average performance of PSEs with data augmentation across different configurations. We plot the median performance across 100 runs. Each tile in the grid represents a different task (obstacle position/floor height combination). For each grid configuration, the height varies along the $y$-axis (11 heights) while the obstacle position varies along the $x$-axis (26 locations). The red letter ⊤ indicates the training tasks. Random grid depicts only one instance, each run consisted of a different test/train split. Beige tiles are tasks PSEs solved while **black** tiles are tasks PSEs did not solve when used with data augmentation. These results were chosen across all the 100 runs to demonstrate what the average reported performance looks like.

precisely, at a specific distance from the obstacle, otherwise it will eventually hit the obstacle. Different tasks consist in shifting the floor height and/or the obstacle position. To generalize, the agent needs to be invariant to the floor height while jump based on the obstacle position. The obstacle can be in 26 different locations while the floor has 11 different heights, totaling 286 tasks.

**Problem Setup**. We split the problem into 18 seen (training) and 268 unseen (test) tasks to stress test generalization using a few changes in the underlying factors of variations seen during training. The small number of positive examples[3] results in a highly unbalanced classification problem with low amounts of data, making it challenging without additional inductive biases. Thus, we evaluate generalization in regimes with and without data augmentation. The different grids configurations (Figure 2) capture different types of generalization: the "wide" grid tests generalization via "interpolation", the "narrow" grid tests out-of-distribution generalization via "extrapolation", and the random grid instances evaluate generalization similar to supervised learning where train and test samples are drawn i.i.d. from the same distribution.

We used RandConv (Lee et al., 2020a), a state-of-the-art data augmentation for generalization. For hyperparameter selection, we evaluate all agents on a validation set containing 54 unseen tasks in the "wide" grid (Figure 2a) and pick the parameters with the best validation performance. We use these fixed parameters for all grid configurations to show the robustness of PSEs to hyperparameter tuning. We first compute the optimal trajectories in the training tasks. Using these trajectories, we compute PSM using dynamic programming (Section D.1). We train the agent by imitation learning, combined with an auxiliary loss for PSEs (Section 4). More details are in Section G.

## 5.1 EVALUATING GENERALIZATION ON JUMPING TASK

We show the efficacy of PSEs compared to common generalization approaches such as regularization (*e.g.,* Cobbe et al., 2019; Farebrother et al., 2018), and data augmentation (*e.g.,* Lee et al., 2020a; Laskin et al., 2020a), which are quite effective on pixel-based RL tasks. We also contrast PSEs with bisimulation transfer (Castro & Precup, 2010), a tabular state-based transfer approach based on bisimulation metrics which does not do any learning and bisimulation preserving representations (Zhang et al., 2021), showing the advantages of PSM over a prevalent state similarity metric.

---

[3]We have 18 different trajectories with several examples for the action *right*, but only one instance of *jump* action per trajectory, leading to just 18 total instances of the action *jump*.

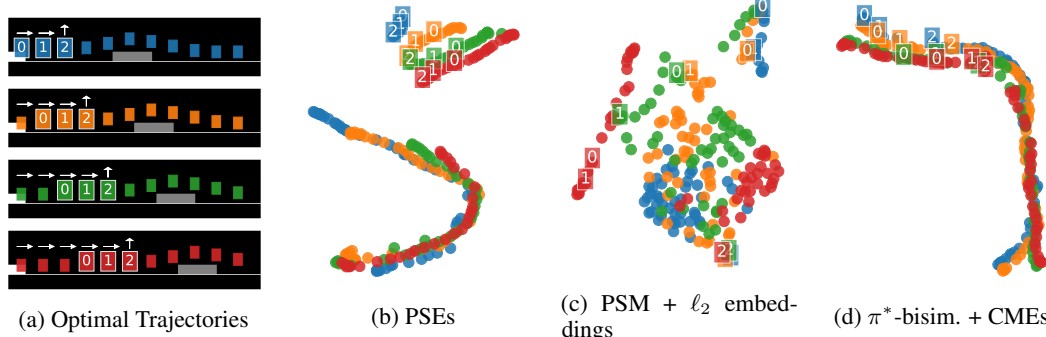

(a) Optimal Trajectories  (b) PSEs  (c) PSM + $\ell_2$ embeddings  (d) $\pi^*$-bisim. + CMEs

Figure 3: **Embedding visualization.** (a) Optimal trajectories on original jumping task (visualized as coloured blocks) with different obstacle positions. We visualize the hidden representations using UMAP, where the color of points indicate the tasks of the corresponding observations. Points with the same number label correspond to same distance of the agent from the obstacle, the underlying optimal invariant feature across tasks.

We first investigated how well PSEs generalize over existing methods without incorporating additional domain knowledge during training. Table 1 summarizes, in the setting without data augmentation, the performance of these methods in different train/test splits (*c.f.* Figure 2 for a detailed description). PSEs, with only 18 examples, already leads to better performance than standard regularization.

PSEs also outperform bisimulation transfer in the "wide" and random grids. Although bisimulation transfer is **impractical**[4] when evaluating zero-shot generalization, we still performed this comparison, unfair to PSEs, to highlight their efficacy. PSEs perform better because, in contrast to bisimulation, PSM is reward agnostic (*c.f.* Proposition D.1) – the expected return of the jump action is quite different depending on the obstacle position (*c.f.* Figure G.2 for a visual juxtaposition of PSM and bisimulation). Overall, these results are promising because they place PSEs as an effective generalization method that does not rely on data augmentation.

Nevertheless, PSEs are complementary to data augmentation, which consistently improves generalization in deep RL. We compared RandConv combined with PSEs to simply using RandConv. Domain-specific augmentation also succeeds in the jumping task. Thus, it is not surprising that RandConv is so effective compared to techniques without augmentation. Table 1 ($2^{nd}$ row) shows that PSEs substantially improve the performance of RandConv across all grid configurations. Moreover, Table 1 ($2^{nd}$ row) illustrates that when combined with RandConv, bisimulation preserving representations (Zhang et al., 2021) diminish generalization by $30 - 50\%$ relative to PSEs.

Notably, Table 1 ($1^{st}$ row) indicates that learning-based methods are ineffective on the "narrow" grid without data augmentation. That said, PSEs do work quite well when combined with RandConv. However, even with data augmentation, generalization in "narrow" grid happens only around the vicinity of training tasks, exhibiting the challenge this grid poses for learning-based methods. We believe this is due to the poor extrapolation ability of neural networks (*e.g.,* Haley & Soloway, 1992; Xu et al., 2020), which is more perceptible without prior inductive bias from data augmentation.

## 5.2 Understanding gains from PSEs: Ablations and Visualizations

PSEs are contrastive metric embeddings (CMEs) learned with PSM. We investigate the gains from CMEs (Section 4) and PSM (Section 3) by ablating them. CMEs can be learned with any state similarity metric – we use $\pi^*$-bisimulation (Castro, 2020) as an alternative. Similarly, PSM can be used with any metric embedding – we use $\ell_2$-embeddings (Section E) as an alternative, which Zhang et al.

Table 2: **Ablating PSEs**. Percentage (%) of test tasks solved when we ablate the similarity metric and learning procedure for metric embeddings in the data augmentation setting on wide grid. PSEs, which combine CMEs with PSM, considerably outperform other embeddings. We report the average performance across 100 runs with standard deviation between parentheses. All ablations except PSEs deteriorate performance compared to just using data augmentation (RandConv), as reported in Table 1.

| Metric / Embedding | $\ell_2$-embeddings | CMEs |
|---|---|---|
| $\pi^*$-bisimulation | 41.4 (17.6) | 23.1 (7.6) |
| PSM | 17.5 (8.4) | **87.0** (10.1) |

[4]Bisimulation transfer assumes oracle access to dynamics and rewards on unseen environments as well as tabular state space to compute the exact bisimulation metric (Section C).

(2021) employed with $\pi^*$-bisimulation for learning representations in a single task RL setting. For a fair comparison, we tune hyperparameters for 128 trials for each ablation entry in Table 2.

Table 2 show that PSEs (= PSM + CMEs) generalize significantly better than $\pi^*$-bisimulation with CMEs or $\ell_2$-embeddings, both of which significantly degrade performance ($-60\%$ and $-45\%$, respectively). This is expected, as $\pi^*$-bisimulation imposes the incorrect invariances for the jumping task (*c.f.* Figures G.2a and G.2d). Additionally, looking at the rows of Table 2, CMEs are superior than $\ell_2$-embeddings for PSM while inferior for $\pi^*$-bisimulation. This result is in line with the hypothesis that CMEs better enforce the invariances encoded by a similarity metric compared to $\ell_2$-embeddings (*c.f.* Figures 3b and 3c).

**Visualizing learned representations**. We visualize the metric embeddings in the ablation above by projecting them to two dimensions with UMAP (McInnes et al., 2018), a popular visualization technique for high dimensional data which better preserves the data's global structure compared to other methods such as t-SNE (Coenen & Pearce, 2019).

Figure 3 shows that PSEs partition the states into two sets: (1) states where a single suboptimal action leads to failure (all states before *jump*) and (2) states where actions do not affect the final outcome (states after *jump*). Additionally, PSEs align the labeled states in the first set, which have a PSM distance of zero. These aligned states have the same distance from the obstacle, the invariant feature that generalizes across tasks. On the other hand, $\ell_2$-embeddings (Zhang et al., 2021) with PSM do not align states with zero PSM except the state with the jump action – poor generalization, as observed empirically, is likely due to states with the same optimal behavior ending up with distant embeddings. CMEs with $\pi^*$-bisimulation align states with $\pi^*$-bisimulation distance of zero – such states are equidistant from the start state and have different optimal behavior for any pair of tasks with different obstacle positions (Figure G.2c).

### 5.3 JUMPING TASK WITH COLORS: WHERE TASK-DEPENDENT INVARIANCE MATTERS

PSEs capture invariances that are usually orthogonal to task-agnostic invariances from data augmentation. This difference is important because, for certain tasks, data augmentation can erroneously alias states with different optimal behavior. Domain knowledge is often required to select appropriate augmentations, otherwise augmentations can hurt generalization. In contrast, PSEs do not require any domain knowledge but instead exploit the inherent structure of the RL tasks.

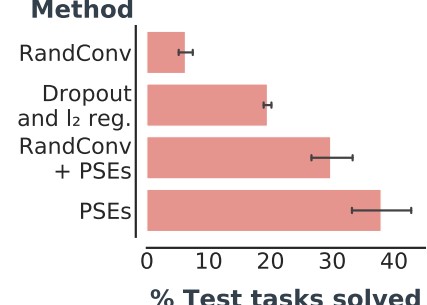

Figure 4: Percentage (%) of red obstacle test tasks solved when trained, jointly with red and green obstacles, on the "wide" grid. We report the mean across 100 runs. Error bars show 99% confidence interval for the mean.

To demonstrate the difference between PSEs and data augmentation, we simply include colored obstacles in the jumping task (see Figure G.5). In this modified task, the optimal behavior of the agent depends on the obstacle color: the agent needs to jump over the red obstacle but strike the green obstacle to get a high return. The red obstacle task has the same difficulty as the original jumping task while the green obstacle task is easier. We jointly train the agent with 18 training tasks each, for both obstacle colors, and evaluate generalization on unseen red tasks.

Figure 4 shows the large performance gap between PSEs and data augmentation with RandConv. All methods solve the green obstacle tasks (Table G.1). As opposed to the original jumping task (*c.f.* Table 1), data augmentation inhibits generalization since RandConv forces the agent to ignore color, conflating the red and green tasks (Figure G.6). PSEs still outperform regularization and data augmentation. Furthermore, data augmentation performs better when combined with PSEs. Thus, PSEs are effective even when data augmentation hurts performance.

### 5.4 EFFECT OF POLICY SUBOPTIMALITY ON PSEs

To understand the sensitivity of learning effective PSEs to the quality of the policies, we compute PSEs using $\epsilon$-suboptimal policies on the jumping task, which take the optimal action with probability $1 - \epsilon$ and the subopotimal action with probability $\epsilon$. We evaluate the generalization performance of PSEs for increasingly suboptimal policies, ranging from the optimal policy ($\epsilon = 0$) to the uniform random policy ($\epsilon = 0.5$). To isolate the effect of suboptimality on PSEs, the agent still imitates the optimal actions during training for all $\epsilon$.

Figure 5 shows that PSEs show near-optimal generalization with $\epsilon \leq 0.4$ while degrade generalization with an uniform random policy. This result is well-aligned with Proposition D.3, which shows that for policies with decreasing suboptimality, PSM approximation becomes more accurate, resulting in improved PSEs. Overall, this study confirms that the utility of PSEs for generalization is robust to suboptimality. One reason for this robustness is that PSEs are likely to align states with similar long-term greedy optimal actions, resulting in good performance even with suboptimal policies that preserve these greedy actions.

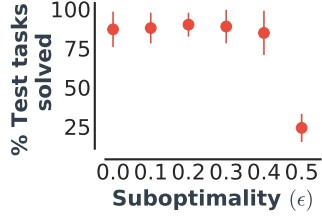

Figure 5: % of test tasks solved using PSEs computed using $\epsilon$-suboptimal policies on the "wide" grid. We report the mean across 100 runs. Error bars show one standard deviation.

## 6 ADDITIONAL EMPIRICAL EVALUATION

In this section, we exhibit that PSM ignores spurious information for generalization using a LQR task (Song et al., 2019) with non-image inputs. Then, we demonstrate the scalability of PSEs without explicit access to optimal policies in an RL setting, with continuous actions, using Distracting DM Control Suite (Stone et al., 2021; Zhang et al., 2018b).

### 6.1 LQR WITH SPURIOUS CORRELATIONS

We show how representations learned using PSM, when faced with semantically equivalent environments, can learn the main factors of variation and ignore spurious correlations that hinder generalization. We use LQR with distractors (Song et al., 2019; Sonar et al., 2020) to assess generalization in a feature-based RL setting with linear function approximation. The distractors are input features that are spuriously correlated with optimal actions and can be used for predicting these actions during training, but hurt generalization. The agent learns a linear policy using 2 environments with fixed distractors. This policy is evaluated on environments with unseen distractors.

We aggregate state pairs in training environments with near-zero PSM. We contrast this approach with (i) IPO (Sonar et al., 2020), a policy optimization method based on IRM (Arjovsky et al., 2019) for out-of-distribution generalization, (ii) overparametrization,

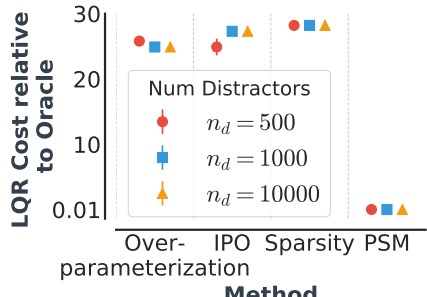

Figure 6: **LQR generalization**: Absolute test error in LQR cost relative to the oracle (which has access to true state), of various methods trained with $n_d$ distractors on 2 training environments. Lower error is better. PSM achieves near-optimal performance. We report mean across 100 different seeds. Error bars (non-existent for most methods) show 99% confidence interval for mean.

which leads to better generalization via implicit regularization (Song et al., 2019), and (iii) weight sparsity using $\ell_1$-regularization since the policy weights which generalize in this task are sparse.

All methods optimally solve the training environments; however, the baselines perform abysmally in terms of generalization compared to state aggregation with PSM (Figure 6), indicating their reliance on distractors. PSM obtains near-optimal generalization which we corroborate through this conjecture (Section H.1): *Assuming zero state aggregation error with PSM, the policy learned using gradient descent is independent of the distractors*. Refer to Section H for a detailed discussion.

### 6.2 DISTRACTING DM CONTROL SUITE

Finally, we demonstrate scalability of PSEs on the Distracting DM Control Suite (DCS) (Stone et al., 2021), which tests whether agents can ignore high-dimensional visual distractors irrelevant to the RL task. Since we do not have access to optimal training policies, we use learned policies as proxy for $\pi^*$ for computing PSM as well as collecting data for optimizing PSEs. Even with this approximation, PSEs outperform state-of-the-art data augmentation. DCS extends DM Control (Tassa et al., 2020) with visual distractions. We use the dynamic background distractions (Stone et al., 2021; Zhang et al., 2018b) where a video is played in the background from a specific

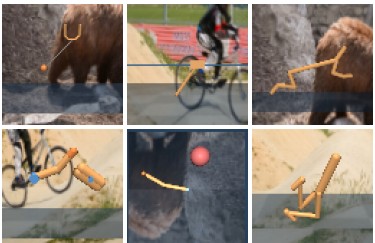

Figure 7: **Distracting Control Suite**: Snapshots of training environments.

frame. The video and the frame are randomly sampled every new episode. We use 2 videos during training (Figure 7) and evaluate generalization on 30 unseen videos (Figure I.1).

Table 3: **Generalization performance** with unseen distractions in the Distracting Suite at 500K steps. We report the average scores across 5 seeds ± standard error. All methods are added to SAC (Haarnoja et al., 2018). Pretrained initialization uses DrQ trained for 500K steps. Figures I.2 and I.3 show generalization curves.

| Initialization | Method | BiC-catch | C-swingup | C-run | F-spin | R-easy | W-walk |
|---|---|---|---|---|---|---|---|
| Random | DrQ | 747±28 | 582±42 | 220±12 | 646±54 | **931±14** | 549±83 |
| | DrQ + PSEs | **821±17** | **749±19** | **308±12** | **779±49** | **955±10** | **789±28** |
| Pretrained | DrQ | 748±30 | 689±22 | 219±10 | **764±48** | 943±10 | 709±29 |
| | DrQ + PSEs | **805±25** | **753±13** | **282±8** | **803±19** | **962±11** | **829±21** |

All agents are built on top of SAC (Haarnoja et al., 2018) combined with DrQ (Kostrikov et al., 2020), an augmentation method with state-of-the-art performance on DM control. Notably, DrQ outperforms CURL (Laskin et al., 2020b), a strong contrastive method for RL. Without data augmentation on DM control, SAC performs poorly, even during training (Kostrikov et al., 2020). We augment DrQ with an auxiliary loss for learning PSEs and compare it with DrQ (Table 3). Orthogonal to DrQ, PSEs align representations of different states across environments based on PSM (*c.f.* Figure A.1). All agents are trained for 500K environment steps with random crop augmentation. For computing PSM, we use policies learned by DrQ pretrained on training environments for 500K steps.

First, we investigate how much better PSEs generalize relative to DrQ, assuming the agent is provided with PSM beforehand. The agent's policy is randomly initialized so that additional gains over DrQ can be attributed to the auxiliary information from PSM. The substantial gains in Table 3 indicate that PSEs are more effective than DrQ for encoding invariance to distractors.

Since PSEs utilize PSM approximated using pretrained policies, we also compare to a DrQ agent where we initialize it using these pretrained policies. This comparison provides the same auxiliary information to DrQ as available to PSEs, thus, the generalization difference stems from how they utilize this information. Table 3 demonstrates that PSEs outperform DrQ with pretrained initialization, indicating that the additional pretraining steps are more judiciously utilized for computing PSM as opposed to just longer training with DrQ. More details, including learning curves, are in Section I.

# 7 RELATED WORK

PSM (Section 3) is inspired by bisimulation metrics (Section C). However, different than traditional bisimulation (*e.g.,* Larsen & Skou, 1991; Givan et al., 2003; Ferns et al., 2011), PSM is more tractable as it defined with respect to a single policy similar to the recently proposed $\pi^*$-bisimulation (Castro, 2020; Zhang et al., 2021). However, in contrast to PSM, bisimulation metrics rely on reward information and may not provide a meaningful notion of behavioral similarity in certain environments (Section 5). For example, states similar under PSM would have similar optimal policies, yet can have arbitrarily large $\pi^*$-bisimulation distance between them (Proposition D.1).

PSEs (Section 4) use contrastive learning to encode behavior similarity (Section 3) *across* MDPs. Previously, contrastive learning has been applied for imposing state self-consistency (Laskin et al., 2020b), capturing predictive information (Oord et al., 2018; Mazoure et al., 2020; Lee et al., 2020b) or encoding transition dynamics (van der Pol et al., 2020; Stooke et al., 2020; Schwarzer et al., 2020) *within* an MDP. These methods can be integrated with PSEs to encode additional invariances. Interestingly, in a similar spirit to PSEs, Pacchiano et al. (2020); Moskovitz et al. (2021) explore comparing behavioral similarity *between policies* to guide policy optimization within an MDP.

PSEs are complementary to data augmentation methods (Kostrikov et al., 2020; Lee et al., 2020a; Raileanu et al., 2020; Ye et al., 2020), which have recently been shown to significantly improve agents' generalization capabilities. In fact, we combine PSEs to state-of-the-art augmentation methods including random convolutions (Lee et al., 2020a; Laskin et al., 2020a) in the jumping task and DrQ (Kostrikov et al., 2020) on Distracting Control Suite, leading to performance improvement.

# 8 CONCLUSION

This paper advances generalization in RL by two contributions: (1) the *policy similarity metric* (PSM) which provides a new notion of state similarity based on behavior proximity, and (2) *contrastive metric embeddings*, which harness the benefits of contrastive learning for representations based on a similarity metric. PSEs combine these two ideas to improve generalization. Overall, this paper shows the benefits of exploiting the inherent structure in RL for learning effective representations.

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

# Appendix

## A  LEARNING CONTRASTIVE METRIC EMBEDDINGS

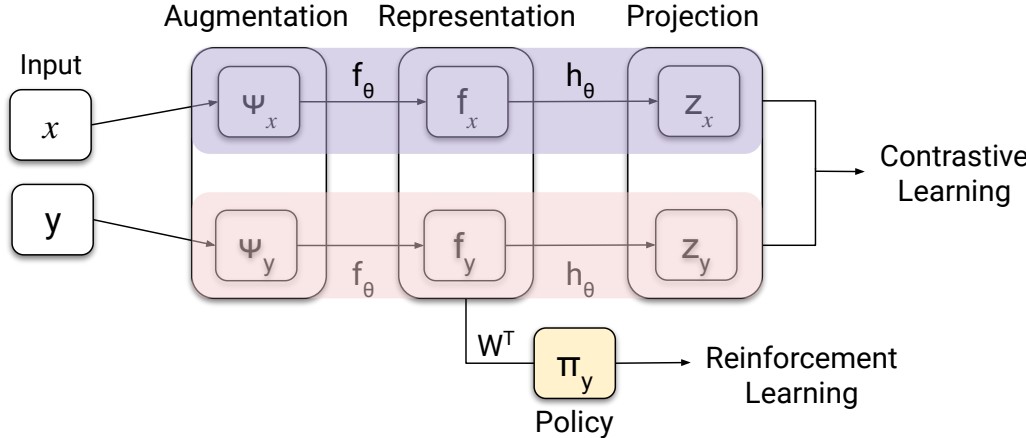

Figure A.1: **Architecture for learning CMEs**. Given an input pair $(x, y)$, we first apply the (optional) data augmentation operator $\Psi$ to produce the input *augmentations* $\Psi_x := \Psi(x), \Psi_y := \Psi(y)$. When not using data augmentation, $\Psi$ is equal to the identity operator, that is, $\forall x \; \Psi(x) = x$. The agent's policy network then outputs the *representations* for these augmentations by applying the encoder $f_\theta$, that is, $f_x = f_\theta(\Psi_x)$, $f_y = f_\theta(\Psi_y)$. These representations are projected using a non-linear projector $h_\theta$ to obtain the embedding $z_\theta$, that is, $z_\theta(x) = h_\theta(f_x)$, $z_\theta(y) = h_\theta(f_y)$. These metric embeddings are trained using the contrastive loss defined in Equation (4). The policy $\pi_\theta$ is an affine function of the representation, that is, $\pi_\theta(\cdot|y) = W^T f_y + b$, where $W, b$ are learned weights and biases. The entire network is trained end-to-end jointly using the reinforcement learning (or imitation learning) loss in conjunction with the auxiliary contrastive loss.

## B  PROOFS

We begin by defining some notation which will be used throughout these results:

- We denote $\mathbb{E}_{t \geq 0}[\gamma^t TV(\tilde{\pi}(Y_y^t), \pi^*(Y_y^t))] = \mathbb{E}_{Y_y^t}\left[\sum_{t \geq 0} \gamma^t TV(\tilde{\pi}(Y_y^t), \pi^*(Y_y^t))\right]$

- For any $y \in Y$, let $Y_y^t \sim P^{\tilde{\pi}}(\cdot|Y_y^{t-1})$, where $Y_y^0 = y$.

- $TV^n(Y_y^k) = \mathbb{E}_{0 \leq t < n}\gamma^t TV(\tilde{\pi}(Y_y^{k+t}), \pi^*(Y_y^{k+t}))$.

We now proceed with some technical lemmas necessary for the main result.

**Lemma 1.** *Given any two pseudometrics[5] $d, d' \in \mathbb{M}$ and probability distributions $P_\mathcal{X}, P_\mathcal{Y}$ where $\mathcal{X}, \mathcal{Y} \subset \mathcal{S}$, we have:*

$$\mathcal{W}^1(d)(P_\mathcal{X}, P_\mathcal{Y}) \leq \|d - d'\| + \mathcal{W}^1(d')(P_\mathcal{X}, P_\mathcal{Y})$$

*Proof.* Note that the dual of the linear program for computing $\mathcal{W}^1(d)(P_\mathcal{X}, P_\mathcal{Y})$ is given by

$$\min_\Gamma \sum_{x \in \mathcal{X},\, y \in \mathcal{Y}} \Gamma(x, y)\, d(x, y)$$

$$\text{subject to } \sum_x \Gamma(x, y) = P_\mathcal{Y}(y) \; \forall y, \quad \sum_y \Gamma(x, y) = P_\mathcal{X}(x) \; \forall x, \quad \Gamma(x, y) \geq 0 \; \forall x, y$$

---

[5]Pseudometrics are generalization of metrics where the distance between two distinct states can be zero.

Using the dual formulation subject to the constraints above, $\mathcal{W}^1(d)$ can be written as

$$
\begin{aligned}
\mathcal{W}^1(d)(P_{\mathcal{X}}, P_{\mathcal{Y}}) \leq \|d - d'\| = \mathcal{W}^1(d - d' + d')(P_{\mathcal{X}}, P_{\mathcal{Y}}) &\leq \|d - d'\| \\
&= \min_{\Gamma} \sum_{x \in \mathcal{X}, \, y \in \mathcal{Y}} \Gamma(x, y) \left( d(x, y) - d'(x, y) + d'(x, y) \right) \\
&\leq \min_{\Gamma} \sum_{x \in \mathcal{X}, \, y \in \mathcal{Y}} \Gamma(x, y) \left( \|d - d'\| + d'(x, y) \right) \\
&= \|d - d'\| + \min_{\Gamma} \sum_{x \in \mathcal{X}, \, y \in \mathcal{Y}} \Gamma(x, y) \, d'(x, y) \\
&= \|d - d'\| + \mathcal{W}^1(d')(P_{\mathcal{X}}, P_{\mathcal{Y}})
\end{aligned}
$$

$\square$

**Lemma 2.** *Given any $y_0 \in Y$, we have:*

$$
\sum_{y_1 \in Y} \left( P^{\tilde{\pi}}(y_1|y_0) - P^{\pi^*}(y_1|y_0) \right) TV^n(Y_{y_1}^0) \leq \frac{2}{1 - \gamma} TV(\tilde{\pi}(y_0), \pi^*(y_0))
$$

*Proof.*

$$
\begin{aligned}
\sum_{y_1 \in Y} \left( P^{\tilde{\pi}}(y_1|y_0) - P^{\pi^*}(y_1|y_0) \right) TV^n(Y_{y_1}^0) &\leq \left| \sum_{y_1 \in Y} \left( P^{\tilde{\pi}}(y_1|y_0) - P^{\pi^*}(y_1|y_0) \right) TV^n(Y_{y_1}^0) \right| \\
&\leq \sum_{y_1 \in Y} \left| \sum_{a \in A} P(y_1|y_0, a) \left( \tilde{\pi}(a|y_0) - \pi^*(a|y_0) \right) \right| TV^n(Y_{y_1}^0) \\
&\leq \frac{1}{1 - \gamma} \sum_{y_1 \in Y} \sum_{a \in A} P(y_1|y_0, a) \left| \tilde{\pi}(a|y_0) - \pi^*(a|y_0) \right| \\
&= \frac{1}{1 - \gamma} \sum_{a \in A} \left| \tilde{\pi}(a|y_0) - \pi^*(a|y_0) \right| \sum_{y_1 \in Y} P(y_1|y_0, a) \\
&= \frac{1}{1 - \gamma} \sum_{a \in A} \left| \tilde{\pi}(a|y_0) - \pi^*(a|y_0) \right| \\
&= \frac{2}{1 - \gamma} TV(\tilde{\pi}(y_0), \pi^*(y_0))
\end{aligned}
$$

$\square$

**Lemma 3.** *Given any $y_0 \in Y$, if $TV^n(Y_{y_1}^0) \leq \frac{1+\gamma}{1-\gamma} d^*(\tilde{x}_{y_1}, y_1)$, we have:*

$$
\sum_{y_1 \in Y} P^{\pi^*}(y_1|y_0) TV^n(Y_{y_1}^0) \leq \frac{1 + \gamma}{1 - \gamma} W_1(d^*) \left( P^{\pi^*}(\cdot|\tilde{x}_{y_0}), P^{\pi^*}(\cdot|y_0) \right)
$$

*Proof.* Note that we have the following equality, where $\mathbf{0}$ is a vector of zeros:

$$
\sum_{y_1 \in Y} P^{\pi^*}(y_1|y_0) TV^n(Y_{y_1}^0) = \sum_{y_1 \in Y} P^{\pi^*}(y_1|y_0) TV^n(Y_{y_1}^0) - \sum_{x \in X} P^{\pi^*}(x|\tilde{x}_{y_0}) \mathbf{0}
$$

which is the same form as the primal LP for $W_1(d^*)(P^{\pi^*}(\cdot|y_0), P^{\pi^*}(\cdot|\tilde{x}_{y_0}))$. By assumption, we have that

$$
TV^n(Y_{y_1}^0) \leq \frac{1 + \gamma}{1 - \gamma} d^*(\tilde{x}_{y_1}, y_1)
$$

This implies that $\frac{1-\gamma}{1+\gamma} TV^n(Y_{\cdot}^0)$ is a feasible solution to $W_1(d^*)(P^{\pi^*}(\cdot|y_0), P^{\pi^*}(\cdot|\tilde{x}_{y_0}))$:

$$
\sum_{y_1 \in Y} P^{\pi^*}(y_1|y_0) \frac{1 - \gamma}{1 + \gamma} TV^n(Y_{y_1}^0) \leq W_1(d^*) \left( P^{\pi^*}(\cdot|\tilde{x}_{y_0}), P^{\pi^*}(\cdot|y_0) \right)
$$

and the result follows. $\square$

**Proposition B.1.** *The operator $\mathcal{F}$ given by:*

$$\mathcal{F}(d)(x,y) = \text{DIST}(\pi^*(x), \pi^*(y)) + \gamma \mathcal{W}_1(d)(P_{\mathcal{X}}^{\pi^*}(\cdot|x), P_{\mathcal{Y}}^{\pi^*}(\cdot|y))$$

*is a contraction mapping and has a unique fixed point for a bounded $dist$.*

*Proof.* We first prove that $\mathcal{F}$ is contraction mapping. Then, a simple application of the Banach Fixed Point Theorem asserts that $\mathcal{F}$ has a unique fixed point. Note that for all pseudometrics $d, d' \in \mathbb{M}$, and for all states $x \in \mathcal{X}$, $y \in \mathcal{Y}$,

$$\mathcal{F}(d)(x,y) - \mathcal{F}(d')(x,y)$$
$$= \gamma \left( \mathcal{W}_1(d)(P^{\pi^*_{\mathcal{X}}}(\cdot|x), P_{\mathcal{Y}}^{\pi^*}(\cdot|y)) - \mathcal{W}_1(d')(P^{\pi^*_{\mathcal{X}}}(\cdot|x), P_{\mathcal{Y}}^{\pi^*}(\cdot|y)) \right)$$
$$\overset{\text{Lemma 1}}{\leq} \gamma \left( \|d - d'\| + \mathcal{W}_1(d')(P^{\pi^*_{\mathcal{X}}}(\cdot|x), P_{\mathcal{Y}}^{\pi^*}(\cdot|y)) - \mathcal{W}_1(d')(P^{\pi^*_{\mathcal{X}}}(\cdot|x), P_{\mathcal{Y}}^{\pi^*}(\cdot|y)) \right)$$
$$= \gamma \|d - d'\|$$

Thus, $\|\mathcal{F}(d) - \mathcal{F}(d')\| \leq \gamma \|d - d'\|$, so that $\mathcal{F}$ is a contracting mapping for $\gamma < 1$ and has an unique fixed point $d^*$. $\qquad\square$

**Theorem 1.** *[Bound on policy transfer] For any $y \in \mathcal{Y}$, let $Y_y^t \sim P^{\tilde{\pi}}(\cdot \,|\, Y_y^{t-1})$ define the sequence of random states encountered starting in $Y_y^0 = y$ and following policy $\tilde{\pi}$. We have:*

$$\mathbb{E}_{Y_y^t} \left[ \sum_{t \geq 0} \gamma^t TV\left(\tilde{\pi}(Y_y^t), \pi^*(Y_y^t)\right) \right] \leq \frac{1+\gamma}{1-\gamma} d^*(\tilde{x}_y, y) \,.$$

*Proof.* We will prove this by induction. Assuming that the bound holds for $TV^n$, we prove the bound holds for $TV^{n+1}$. The base case for $n = 1$ follows from $TV(\tilde{\pi}(y), \pi^*(y)) = TV(\pi^*(\tilde{x}_y), \pi^*(y)) \leq d^*(\tilde{x}_y, y)$. Note that $TV^n \leq \frac{1-\gamma^{n+1}}{1-\gamma}$ since the $TV$ distance per time-step can be at most 1.

Let $P_t^\pi(y'|y)$ denote the probability of ending in state $y' \in Y$ after $t$ steps when following policy $\pi$ and starting from state $y$. We then have:

$$TV^{n+1}(Y_y^k) = \sum_{y_k \in Y} P_k^{\tilde{\pi}}(y_k|y) TV(\tilde{\pi}(y_k), \pi^*(y_k)) + \gamma TV^n(Y_y^{k+1})$$

$$= \sum_{y_k \in Y} P_k^{\tilde{\pi}}(y_k|y) \left[ TV(\tilde{\pi}(y_k), \pi^*(y_k)) + \gamma \sum_{y_{k+1} \in Y} P^{\tilde{\pi}}(y_{k+1}|y_k) TV^n(Y_{y_{k+1}}^0) \right]$$

$$= \sum_{y_k \in Y} P_k^{\tilde{\pi}}(y_k|y) \left[ TV(\tilde{\pi}(y_k), \pi^*(y_k)) + \gamma \sum_{y_{k+1} \in Y} \left( P^{\tilde{\pi}}(y_{k+1}|y_k) - P^{\pi^*}(y_{k+1}|y_k) \right) TV^n(Y_{y_{k+1}}^0) \right.$$

$$\left. + \gamma \sum_{y_{k+1} \in Y} P^{\pi^*}(y_{k+1}|y_k) TV^n(Y_{y_{k+1}}^0) \right]$$

$$\overset{\text{Lemma 2}}{\leq} \sum_{y_k \in Y} P_k^{\tilde{\pi}}(y_k|y) \left[ TV(\tilde{\pi}(y_k), \pi^*(y_k)) + \frac{2\gamma}{1-\gamma} TV(\tilde{\pi}(y_k), \pi^*(y_k)) + \gamma \sum_{y_{k+1} \in Y} P^{\pi^*}(y_{k+1}|y_k) TV^n(Y_{y_{k+1}}^0) \right]$$

$$\overset{\text{Lemma 3}}{\leq} \sum_{y_k \in Y} P_k^{\tilde{\pi}}(y_k|y) \left[ TV(\tilde{\pi}(y_k), \pi^*(y_k)) \right.$$

$$\left. + \gamma \left( \frac{2}{1-\gamma} TV(\tilde{\pi}(y_k), \pi^*(y_k)) + \frac{1+\gamma}{1-\gamma} W_1(d^*)\left(P^{\pi^*}(\cdot|\tilde{x}_{y_k}), P^{\pi^*}(\cdot|y_k)\right) \right) \right]$$

$$= \sum_{y_k \in Y} P_k^{\tilde{\pi}}(y_k|y) \left[ \frac{1+\gamma}{1-\gamma} \left( TV(\pi^*(\tilde{x}_{y_k}), \pi^*(y_k)) + \gamma W_1(d^*)(P^{\pi^*}(\cdot|\tilde{x}_{y_k}), P^{\pi^*}(\cdot|y_k)) \right) \right]$$

$$\leq \sum_{y_k \in Y} P_k^{\tilde{\pi}}(y_k|y) \frac{1+\gamma}{1-\gamma} d^*(\tilde{x}_{y_k}, y_k)$$

Thus, by induction, it follows that for all $n$:

$$TV^n(Y_y^0) \leq \frac{1+\gamma}{1-\gamma} d^*(\tilde{x}_y, y),$$

which completes the proof. □

## C  Bisimulation Metrics

**Notation**. We use the notation as defined in Section 2.

Bisimulation metrics (Givan et al., 2003; Ferns et al., 2011) define a pseudometric $d_\sim : \mathcal{S} \times \mathcal{S} \to \mathbb{R}$ where $d_\sim(x, y)$ is defined in terms of distance between immediate rewards and next state distributions. Define $\mathcal{F}_\sim^e : \mathbb{M} \to \mathbb{M}$ by

$$\mathcal{F}_\sim^e(d)(x, y) = \max_{a \in \mathcal{A}} |R(x, a) - R(y, a)| + \gamma \mathcal{W}_1(d)\left(P^a(\cdot \,|\, x), P^a(\cdot \,|\, y)\right) \tag{C.1}$$

then, $\mathcal{F}_\sim^e$ has a unique fixed point $d_\sim$ which is a bisimulation metric. $\mathcal{F}_\sim^e$ uses the 1-Wasserstein metric $\mathcal{W}_1 : \mathbb{M} \to \mathbb{M}_p$. The 1-Wasserstein distance $\mathcal{W}_1(d)$ under the pseudometric $d$ can be computed using the dual linear program:

$$\max_{\mathbf{u}, \mathbf{v}} \sum_{x \in \mathcal{X}} P(x)u_x - \sum_{y \in \mathcal{Y}} P(y)v_y \quad \text{subject to } \forall x \in \mathcal{X}, y \in \mathcal{Y} \quad u_x - v_y \leq d(x, y)$$

Since we are only interested in computing the coupling between the states in $\mathcal{X}$ and $\mathcal{Y}$, the above formulation assumes that $P_\mathcal{X}(y) = 0$ for all $y \in \mathcal{Y}$ and $P_\mathcal{Y}(x) = 0$ for all $x \in \mathcal{X}$. The computation of $d_\sim$ is expensive and requires a tabular representation of the states, rendering it impractical for large state spaces. On-policy bisimulation (Castro, 2020) (*e.g.*, $\pi^*$-bisimulation) is tied to specific behavior policies and is much easier to approximate than bisimulation.

## D  Policy Similarity Metric

### D.1  Computing PSM

In general, PSM for a given DIST across MDPs $\mathcal{M}_\mathcal{X}$ and $\mathcal{M}_\mathcal{Y}$ is given by

$$d^*(x, y) = \text{DIST}\left(\pi_\mathcal{X}^*(x), \pi_\mathcal{Y}^*(y)\right) + \gamma \mathcal{W}_1\left(d^*\right)(P_\mathcal{X}^{\pi^*}(\cdot \,|\, x), P_\mathcal{Y}^{\pi^*}(\cdot \,|\, y)). \tag{D.1}$$

Since our main focus is showing the utility of PSM for generalization, we simply use environments where PSM can be computed using dynamic programming. Using a similar observation to Castro (2020), we assert that the recursion for $d^*$ takes the following form in deterministic environments:

$$d^*(x, y) = \text{DIST}\left(\pi_\mathcal{X}^*(x), \pi_\mathcal{Y}^*(y)\right) + \gamma d^*(x', y'). \tag{D.2}$$

where $x' = P_\mathcal{X}^{\pi^*}(x)$, $y' = P_\mathcal{Y}^{\pi^*}(y)$ are the next states from taking actions $\pi_\mathcal{X}^*(x)$, $\pi_\mathcal{X}^*(y)$ from states $x$, $y$ respectively. Furthermore, we assume that DIST between terminal states from $\mathcal{M}_\mathcal{X}$ and $\mathcal{M}_\mathcal{Y}$ is zero. Note that the form of Equation D.2 closely resembles the update rule in Q-learning, and as such, can be efficiently computed with samples using approximate dynamic programming. Given access to optimal trajectories $\tau_\mathcal{X}^* = \{x_t\}_{t=1}^N$ and $\tau_\mathcal{Y}^* = \{y_t\}_{t=1}^N$, where $x_{t+1} = P_\mathcal{X}^{\pi^*}(x_t)$ and $y_{t+1} = P_\mathcal{Y}^{\pi^*}(y_t)$, Equation D.2 can be solved using exact dynamic programming; we provide pseudocode in Section J.1.

There are other ways to approximate the Wasserstein distance in bisimulation metrics (*e.g.,* Ferns et al., 2006; 2011; Castro, 2020; Zhang et al., 2021). That said, approximating bisimulation (or PSM) for stochastic environments remains an exciting research direction (Castro, 2020). Investigating other distance metrics for long-term behavior difference in PSM is also interesting for future work.

### D.2  PSM Connections to Data Augmentation and Bisimulation

**Connection to bisimulation**. Although bisimulation metrics have appealing properties such as bounding value function differences (*e.g.,* (Ferns & Precup, 2014)), they rely on reward information and may not provide a meaningful notion of behavioral similarity in certain environments. Proposition D.1 implies that states similar under PSM would have similar optimal policies yet can have arbitrarily large bisimulation distance between them.

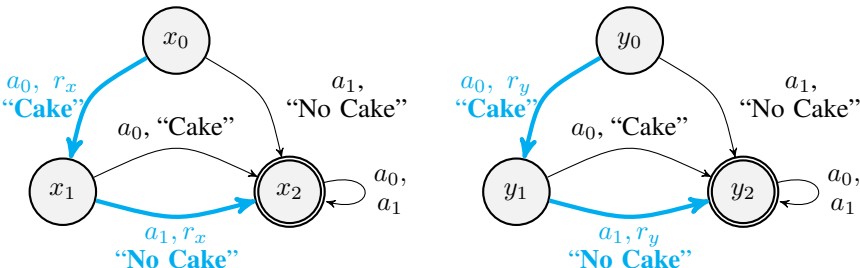

Figure D.1: Cyan edges represent actions with a positive reward, which are also the optimal actions. Zero rewards everywhere else. $x_0, y_0$ are the start states while $x_2, y_2$ are the terminal states.

**Proposition D.1.** *There exists environments $M_{\mathcal{X}}$ and $M_{\mathcal{Y}}$ such that $\forall (x, y) \in \mathcal{L}$ where $\mathcal{L} = \{(x, y) \,|\, x \in \mathcal{X},\ y \in \mathcal{Y},\ d^*(x, y) = 0\}$, $d^*_{\sim}(x, y) = \frac{|R_{\max} - R_{\min}|}{1 - \gamma} - \epsilon$ for any given $\epsilon > 0$.*

For example, consider the two semantically equivalent environments in Figure D.1 with $\pi^*_{\mathcal{X}}(x_0) = \pi^*_{\mathcal{Y}}(y_0) = a_0$ and $\pi^*_{\mathcal{X}}(x_1) = \pi^*_{\mathcal{Y}}(y_1) = a_1$ but different rewards $r_x, r_y$ respectively. Whenever $r_y > (1 + 1/\gamma)\, r_x$, bisimulation metrics incorrectly imply that $x_0$ is more behaviorally similar to $y_1$ than $y_0$.

For the MDPs shown in Figure D.1, to determine which $y$ state is behaviorally equivalent to $x_0$, we look at the distances computed by bisimulation metric $d_{\sim}$ and $\pi^*$-bisimulation metric $d^*_{\sim}$:

$$d_{\sim}(x_0, y_0) = d^*_{\sim}(x_0, y_0) = (1 + \gamma)|r_y - r_x|$$
$$d_{\sim}(x_0, y_1) = \max\left((1 + \gamma)\, r_x, r_y\right),\ d^*_{\sim}(x_0, y_1) = |r_y - r_x| + \gamma r_x$$

Thus, $r_y > (1 + 1/\gamma)\, r_x$ implies that $d_{\sim}(x_0, y_1) < d_{\sim}(x_0, y_0)$ as well as $d^*_{\sim}(x_0, y_1) < d^*_{\sim}(x_0, y_0)$.

**Connection to data augmentation.** Data augmentation often assumes access to optimality invariant transformations, *e.g.,* random crops or flips in image-based benchmarks (Laskin et al., 2020a; Kostrikov et al., 2020). However, for certain RL tasks, such augmentations can erroneously alias states with different optimal behavior and hurt generalization. For example, if the image observation is flipped in a goal reaching task with left and right actions, the optimal actions would also be flipped to take left actions instead of right and vice versa. Proposition D.2 states that PSMs can precisely quantify the invariance of such augmentations.

**Proposition D.2.** *For an MDP $M_{\mathcal{X}}$ and its transformed version $M_{\psi(\mathcal{X})}$ for the data augmentation $\psi$, $d^*(x, \psi(x))$ indicates the optimality invariance of $\psi$ for any $x \in \mathcal{X}$.*

### D.3 PSM with Approximately-Optimal Policies

**Generalized Policy Similarity Metric for arbitrary policies.** For a given DIST, we define a generalized PSM $d : (\mathcal{S} \times \Pi) \times (\mathcal{S} \times \Pi) \to \mathbb{R}$ where $\Pi$ is the set of all policies over $\mathcal{S}$. $d$ satisfies the recursive equation:

$$d\big((x, \pi_1), (y, \pi_2)\big) = \text{DIST}\big(\pi_1(x), \pi_2(y)\big) + \gamma \mathcal{W}_1(d)\big(P^{\pi_1}(\cdot \,|\, x), P^{\pi_2}(\cdot \,|\, y)\big). \tag{D.3}$$

Since DIST is assumed to be a pseudometric and $\mathcal{W}_1$ is a probability metric, it implies that $d$ is a pseudometric as (1) $d$ is non-negative, that is, $d\big((x, \pi_1), (y, \pi_2)\big) \geq 0$, (2) $d$ is symmetric, that is, $d\big((x, \pi_1), (y, \pi_2)\big) = d\big((x, \pi_1), (y, \pi_2)\big)$, and $d$ satisfies the triangle inequality, that is, $d\big((x, \pi_1), (y, \pi_2)\big) < d\big((x, \pi_1), (z, \pi_3)\big) + d\big((z, \pi_3), (y, \pi_2)\big)$.

Using this notion of generalized PSM, we show that the approximation error in PSM from using a suboptimal policy is bounded by the policy's suboptimality. Thus, for policies with decreasing suboptimality, the PSM approximation becomes more accurate, resulting in improved PSEs.

**Proposition D.3.** *[Approximation error in PSM] Let $\hat{d} : \mathcal{S} \times \mathcal{S} \to \mathbb{R}$ be the approximate PSM computed using a suboptimal policy $\hat{\pi}$ defined over $\mathcal{S}$, that is, $\hat{d}(x, y) = \text{DIST}\big(\hat{\pi}(x), \hat{\pi}(y)\big) + \gamma \mathcal{W}_1(\hat{d})\big(P^{\hat{\pi}}(\cdot \,|\, x), P^{\hat{\pi}}(\cdot \,|\, y)\big)$. We have:*

$$|d^*(x, y) - \hat{d}(x, y)| < \underbrace{d\big((x, \pi^*), (x, \hat{\pi})\big)}_{\substack{\text{Long-term suboptimality} \\ \text{difference from } x}} + \underbrace{d\big((y, \hat{\pi}), (y, \pi^*)\big)}_{\substack{\text{Long-term suboptimality} \\ \text{difference from } y}}.$$

*Proof.* The PSM $d^*$ and approximate PSM $\hat{d}$ are instantiations of the generalized PSM (Equation D.3) with both input policies as $\pi^*$ and $\hat{\pi}$ respectively.

$$d^*(x,y) = d\big((x,\pi^*),(y,\pi^*)\big) < d\big((x,\pi^*),(x,\hat{\pi})\big) + d\big((x,\hat{\pi}),(y,\pi^*)\big)$$

$$< d\big((x,\pi^*),(x,\hat{\pi})\big) + d\big((y,\hat{\pi}),(y,\pi^*)\big) + d\big((x,\hat{\pi}),(y,\hat{\pi})\big)$$

$$d^*(x,y) - \hat{d}(x,y) < d\big((x,\pi^*),(x,\hat{\pi})\big) + d\big((y,\hat{\pi}),(y,\pi^*)\big) \quad \because \hat{d}(x,y) = d\big((x,\hat{\pi}),(y,\hat{\pi})\big)$$

$$\text{Similarly, } \hat{d}(x,y) - d^*(x,y) < d\big((x,\pi^*),(x,\hat{\pi})\big) + d\big((y,\hat{\pi}),(y,\pi^*)\big)$$

$\square$

## E    L2 METRIC EMBEDDINGS

Another common choice (Zhang et al., 2021) for learning metric embeddings is to use the squared loss (*i.e., l2*-loss) for matching the euclidean distance between the representations of a pair of states to the metric distance between those states. Concretely, for a given $d^*$ and representation $f_\theta$, the loss $\mathcal{L}(\theta) = \mathbb{E}_{s_i,s_j}[(\|f_\theta(s_i) - f_\theta(s_j)\|_2 - d^*(s_i,s_j))^2]$ is minimized. However, it might be too restrictive to match the exact metric distances, which we demonstrate empirically by comparing $l2$ metric embeddings with CMEs (Section 5.2).

## F    EXTENDED RELATED WORK

Generalization across different tasks used to be described as *transfer learning*. In the past, most transfer learning approaches relied on fixed representations and tackled different problem formulations (e.g., assuming shared state space). Taylor & Stone (2009) present a comprehensive survey of the techniques at the time, before representation learning became so prevalent in RL. Recently, the problem of performing well in a different, but related task, started to be seen as a problem of *generalization*; with the community highlighting that deep RL agents tend to overfit to the environments they are trained on (Cobbe et al., 2019; Witty et al., 2018; Farebrother et al., 2018; Juliani et al., 2019; Kostrikov et al., 2020; Song et al., 2019; Justesen et al., 2018; Packer et al., 2018).

Prior generalization approaches are typically adapted from supervised learning, including data augmentation, regularization (Cobbe et al., 2019; Farebrother et al., 2018), stochasticity (Zhang et al., 2018c), noise injection (Igl et al., 2019; Zhang et al., 2018a), more diverse training conditions (Rajeswaran et al., 2017; Witty et al., 2018) and self-attention architectures (Tang et al., 2020). In contrast, PSEs exploits behavior similarity (Section 3), a property related to the sequential aspect of RL. Furthermore, for certain RL tasks, it can be unclear what an optimality invariant data augmentation would look like (Section 5.3). PSM can quantify the invariance of such augmentations (Proposition D.2).

Meta-learning is also related to generalization. Meta-learning methods try to find a parametrization that requires a small number of gradient steps to achieve good performance on a new task (Finn et al., 2017). In this context, various meta-learning approaches capable of zero-shot generalization have been proposed (Li et al., 2018; Agarwal et al., 2019; Balaji et al., 2018). These approaches typically consist in minimizing the loss in the environments the agent is while adding an auxiliary loss for ensuring improvement in the other (validation) environments available to the agent. Nevertheless, Tachet des Combes et al. (2018) has shown that meta-learning approaches fail in the jumping task which we also observed empirically. Others have also reported similar findings (e.g., Farebrother et al., 2018).

There are several other approaches for tackling zero-shot generalization in RL, but they often rely on domain-specific information. Some examples include knowledge about equivalences between entities in the environment (Oh et al., 2017) and about what is under the agent's control (Ye et al., 2020). Causality-based methods are a different way of tackling generalization, but current solutions do not scale to high-dimensional observation spaces (e.g., Killian et al., 2017; Perez et al., 2020; Zhang et al., 2020).

# G  JUMPING TASK WITH PIXELS

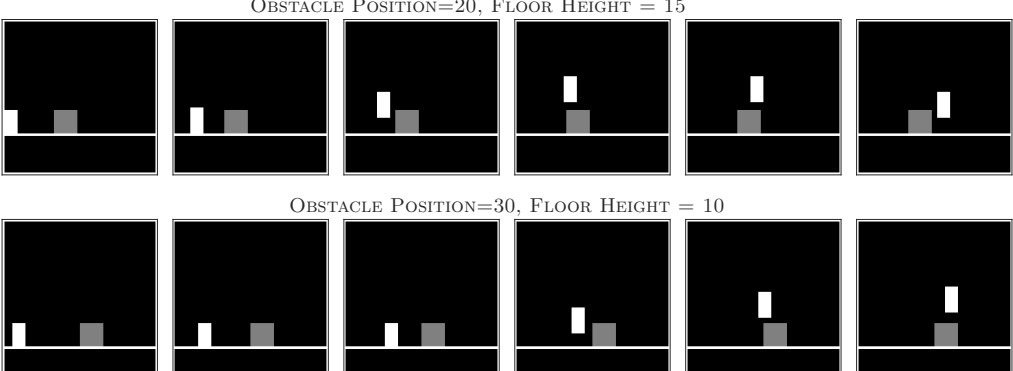

Figure G.1: Optimal trajectories on the jumping tasks for two different environments. Note that the optimal trajectory is a sequence of *right* actions, followed by a single *jump* at a certain distance from the obstacle, followed by *right* actions.

**Detailed Task Description**. The jumping task consists of an agent trying to jump over an obstacle on a floor. The environment is deterministic with the agent observing a reward of $+1$ at each time step. If the agent successfully reaches the rightmost side of the screen, it receives a bonus reward of $+100$; if the agent touches the obstacle, the episode terminates. The observation space is the pixel representation of the environment, as depicted in Figure 1. The agent has access to two actions: *right* and *jump*. The *jump* action moves the agent vertically and horizontally to the right.

**Architecture**. The neural network used for Jumping Task experiment is adapted from the Nature DQN architecture. Specifically, the network consists of 3 convolutional layers of sizes 32, 64, 64 with filter sizes $8 \times 8$, $4 \times 4$ and $3 \times 3$ and strides 4, 2, and 1, respectively. The output of the convnet is fed into a single fully connected layer of size 256 followed by 'ReLU' non-linearity. Finally, this FC layer output is fed into a linear layer which computes the policy which outputs the probability of the *jump* and *right* actions.

**Contrastive Embedding**. For all our experiments, we use a single ReLU layer with $k = 64$ units for the non-linear projection to obtain the embedding $z_\theta$ (Figure A.1). We compute the embedding using the penultimate layer in the jumping task network. Hyperparameters are reported in Table G.2.

**Total Loss**. For jumpy world, the total loss is given by $\mathscr{L}_{\mathrm{IL}} + \alpha\mathscr{L}_{\mathrm{CME}}$ where $\mathscr{L}_{\mathrm{IL}}$ is the imitation learning loss, $\mathscr{L}_{\mathrm{CME}}$ is the auxiliary loss for learning PSEs with the coefficient $\alpha$.

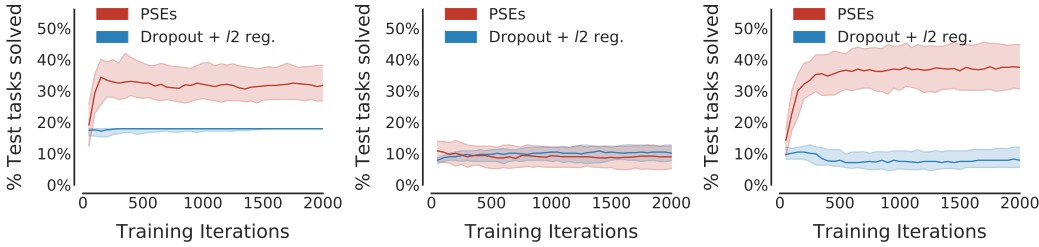

Figure G.3: Test performance curves in the setting **without data augmentation** on the "wide", "narrow", and random grids described in Figure 2. We plot the median performance across 100 runs. Shaded regions show 25 and 75 percentiles.

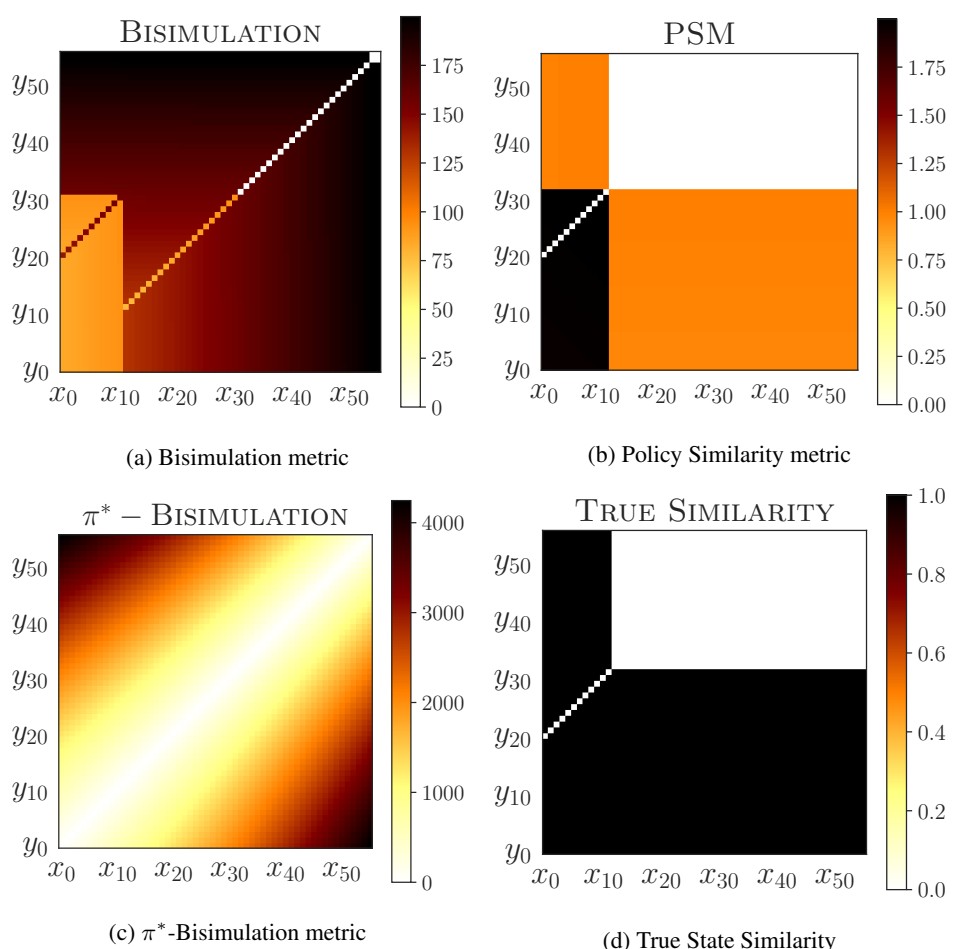

(a) Bisimulation metric

(b) Policy Similarity metric

(c) $\pi^*$-Bisimulation metric

(d) True State Similarity

Figure G.2: **PSM *vs*. Bisimulation**. Visualizing PSM and bisimulation metrics with discount factor $\gamma = 0.99$. $x_i$ and $y_i$ correspond to the states visited by optimal policies in the two environments with obstacle at positions 25 and 45, respectively. For each grid, the $(i, j)^{th}$ location shows the distance assigned by the metric to the states $x_i$ and $y_j$. Lower distances (lighter shades) imply higher similarity between two states. The bottom right figure shows which states are equivalent to each other. The ranges are different for each metric as bisimulation metrics utilize reward differences while PSM uses the total variation $(TV)$ distance between policies. Note that the large bisimulation distances are due to the fact that the reward at the terminal state is set to 100.

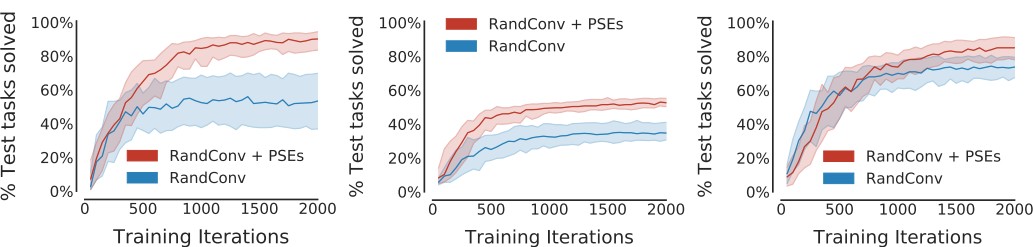

Figure G.4: Test performance curves in the setting **with data augmentation** on the "wide", "narrow", and random grids described in Figure 2. We plot the median performance across 100 runs. Shaded regions show 25 and 75 percentiles.

## G.1 JUMPING TASK FROM COLORS

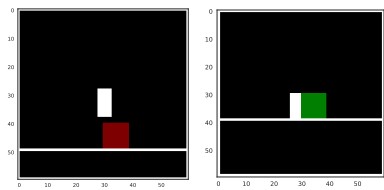

Figure G.5: **Jumping task with colored obstacles**. The agent needs to jump over the red obstacle but strike the green obstacle.

Table G.1: Percentage (%) of test tasks solved when trained on the "wide" grid with both red and green obstacles. The numbers we report are averaged across 100 runs. Standard error is reported between parentheses.

| Method | Red (%) | Green (%) |
|---|---|---|
| RandConv | 6.2 (0.4) | 99.6 (0.2) |
| Dropout and $l_2$ reg. | 19.5 (0.2) | 100.0 (0.0) |
| RandConv + PSEs | 29.8 (1.3) | 99.6 (0.2) |
| PSEs | **37.9** (1.9) | 100.0 (0.0) |

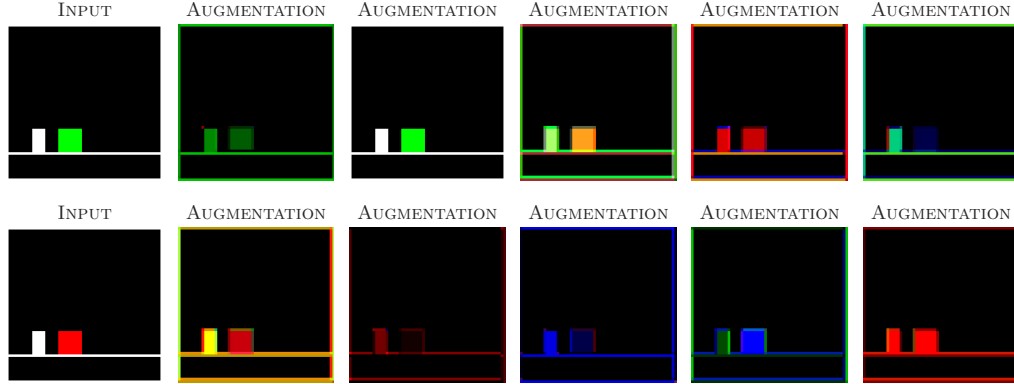

Figure G.6: **Randconv enforces color invariance**. The first column shows the original observation where the top row corresponds to the observation from task with green obstacle jumping task while the bottom row corresponds to the red obstacle jumping task. Columns 2-6 in each row show 5 augmentations by applying RandConv. The augmentations show that RandConv tries to encode invariance with respect to the obstacle color.

## G.2 HYPERPARAMETERS

For hyperparameter selection, we evaluate all agents on a validation set containing 54 unseen tasks in the "wide" grid and pick the parameters with the best validation performance. The validation set (Figure G.7) was selected by using the environments nearby to the training environments whose floor height differ by 1 or whose obstacle position differ by 1.

Table G.2: Common hyperparameters across all methods for all jumping task experiments.

| Hyperparameter | Value |
|---|---|
| Learning rate decay | 0.999 |
| Training epochs | 2000 |
| Optimizer | Adam |
| Batch size (Imitation) | 256 |
| Num training tasks | 18 |
| $\Gamma$-scale Parameter ($\beta$) | 0.01 |
| Embedding size ($k$) | 64 |
| Batch Size ($\mathscr{L}_{\text{CME}}$) | 57 |
| $|\tau_{\mathcal{X}}^*|$ | 57 |

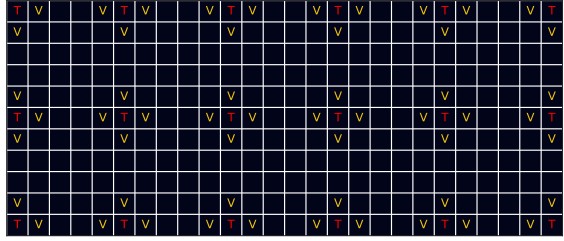

Figure G.7: Unseen 56 validation tasks, labeled with V, used for hyperparameter selection.

Table G.3: Optimal hyperparameters for reporting results in Table 1. These hyperparameters are selected using the "wide" grid by maximizing final performance on a validation set containing 56 unseen tasks. All grid configurations in Table 1 use these hyperparameters.

| Hyperparameter | Dropout and $\ell_2$-reg. | PSEs | RandConv | RandConv + PSEs |
|---|---|---|---|---|
| Learning Rate | $4\times10^{-3}$ | $3.2\times10^{-3}$ | $7\times10^{-3}$ | $2.6\times10^{-3}$ |
| $\ell_2$-reg. coefficient | $4.3\times10^{-4}$ | $1\times10^{-5}$ | – | – |
| Dropout coefficient | $3\times10^{-1}$ | – | – | – |
| Contrastive Temperature $(1/\lambda)$ | – | 1.0 | – | $5\times10^{-1}$ |
| Auxiliary loss coefficient $(\alpha)$ | – | $1\times10^{1}$ | – | 5.0 |

Table G.4: Optimal hyperparameters for reporting results in Figure 5.3. These hyperparameters are selected using the "wide" grid by maximizing final performance on a validation set containing 56 unseen tasks.

| Hyperparameter | Dropout and $\ell_2$-reg. | PSEs | RandConv | RandConv + PSEs |
|---|---|---|---|---|
| Learning Rate | $4\times10^{-3}$ | $6\times10^{-3}$ | $5\times10^{-3}$ | $2.6\times10^{-3}$ |
| $\ell_2$-reg. coefficient | $4.3\times10^{-4}$ | $7\times10^{-5}$ | – | – |
| Dropout coefficient | $3\times10^{-1}$ | – | – | – |
| Contrastive Temperature $(1/\lambda)$ | – | $5\times10^{-1}$ | – | $5\times10^{-1}$ |
| Auxiliary loss coefficient $(\alpha)$ | – | 5.0 | – | 5.0 |

Table G.5: Optimal hyperparameters for reporting ablation results in Table 2. These hyperparameters are selected using the "wide" grid by maximizing final performance on a validation set containing 56 unseen tasks.

| Hyperparameter | PSM | | $\pi^*$-bisimulation | |
|---|---|---|---|---|
| | CMEs | $\ell_2$-embeddings | CMEs | $\ell_2$-embeddings |
| Learning Rate | $4\times10^{-3}$ | $5\times10^{-4}$ | $4.7\times10^{-4}$ | $1\times10^{-4}$ |
| Contrastive Temperature $(1/\lambda)$ | 1.0 | – | $5\times10^{-1}$ | – |
| Auxiliary loss coefficient $(\alpha)$ | 5.0 | $1\times10^{-1}$ | $1\times10^{-1}$ | $1\times10^{-6}$ |

Please note that Table G.3 and Table G.4 correspond to two different tasks: one uses the standard jumping task with white obstacles, while the other uses colored obstacles where the optimal policies depend on color. For fair comparison, we tune hyperparameters for all the methods using Bayesian optimization (Golovin et al., 2017). We use the best parameters among these tuned hyperparameters and the ones found in Table G.3, leading to different parameters for both PSEs as well as RandConv. Evaluating PSEs with the jumping task hyperparameters from Table G.3 instead of the ones in Table G.4 leads to a small drop (-4%) on the jumping task with colors (Section 5.3). Nevertheless, PSEs still outperform other methods in Section 5.3.

## H  LQR: Additional Details

Optimal control with linear dynamics and quadratic cost, commonly known as LQR, has been increasingly used as a simplified surrogate for deep RL problems (Recht, 2019). Following Song et al. (2019); Sonar et al. (2020), we analyze the following LQR problem for assessing generalization:

$$
\begin{aligned}
\text{minimize} \quad & E_{s_0\sim\mathcal{D}}\left[\frac{1}{2}\sum_{t=0}^{\infty} s_t^T Q s_t + a_t^T R a_t\right], \\
\text{subject to} \quad & s_{t+1} = As_t + Ba_t, o_t = \begin{bmatrix} 0.1\ W_c \\ W_d \end{bmatrix} s_t, a_t = Ko_t,
\end{aligned}
\tag{H.1}
$$

where $\mathcal{D}$ is the initial state distribution, $s_t \in \mathbb{R}^{n_s}$ is the (hidden) true state at time t, $a_t \in \mathbb{R}^{n_a}$ is the control action, and $K$ is the linear policy matrix. The agent receives the input observation $o_t$, which is a linear transformation of the state $s_t$. $W_c$ and $W_d$ are semi-orthogonal matrices to prevent information loss for predicting optimal actions. An environment corresponds to a particular choice

Table H.1: **LQR generalization performance**: Absolute error in LQR cost, w.r.t. the oracle solver (which has access to true state), of various methods trained with $n_d$ distractors on $N = 2$ environments. The reported mean and standard deviations are across 100 different seeds. Lower error is better.

| Method | Number of Distractors ($n_d$) | | |
|---|---|---|---|
| | 500 | 1000 | 10000 |
| Overparametrization (Song et al., 2019) | 25.8 (1.5) | 24.9 (1.1) | 24.9 (0.4) |
| IPO (Sonar et al., 2020) (IRM + Policy opt.) | 32.6 (5.0) | 27.3 (2.8) | 24.8 (0.4) |
| Weight Sparsity ($\ell_1$-reg.) | 28.2 (0.0) | 28.2 (0.0) | 28.2 (0.0) |
| PSM (State aggregation) | **0.03** (0.0) | **0.03** (0.0) | **0.02** (0.0) |

of $W_d$; all other system parameters $(A, B, Q, R, W_c)$ are fixed matrices which are shared across environments and unknown to the agent. The agent learns the policy matrix $K$ using $N$ training environments based on Equation H.1. At test time, the learned policy is evaluated on environments with unseen $W_d$.

The generalization challenge in this setup is to ignore the distractors: $W_c s_t \in \mathbb{R}^{n_s}$ represents the state features invariant across environments while $W_d s_t \in \mathbb{R}^{n_d}$ is a high-dimensional distractor of size $n_s, n_d$, respectively, such that $n_s << n_d$. Furthermore, the policy matrix which generalizes across all environments is $K_\star = \begin{bmatrix} 10 \ W_c P_\star^T \\ 0 \end{bmatrix}^T$, where $P_\star$ corresponds to the optimal LQR solution with access to state $s_t$. However, for a single environment with distractor $W_d$, multiple solutions exist, for instance, $K_\star' = \begin{bmatrix} 10\alpha \ W_c P_\star^T \\ (1-\alpha) \ W_d P_\star^T \end{bmatrix}^T \quad \forall \alpha \in [0, 1]$. Note that the distractors are an order of magnitude larger than the invariant features in $o_t$ and dependence on them is likely to cause the agent to act erratically on inputs with unseen distractors, resulting in poor generalization.

We use overparametrized policies with two linear layers, *i.e.,* $K = K_1 K_2$, where $K_1(o)$ is the learned representation for observation $o$. We learn $K$ using gradient descent using the combined cost on 2 training environments with varying number of distractors. We aggregate observation pairs with near-zero PSM by matching their representations using a squared loss. We use the open-source code released by Sonar et al. (2020) for our experiments.

Table H.2: An overview of hyper-parameters for LQR.

| Parameter | Setting |
|---|---|
| A | Orthogonal matrix, scaled 0.8 |
| B | $I_{20\times 20}$ |
| $n_x$ | 20 |
| $n_a$ | 20 |
| Q | $I_{20\times 20}$ |
| R | $I_{20\times 20}$ |
| $K_i \ \forall i$ | Orthogonal Initialization, scaled 0.001 |
| $W_d$ | Random semi-orthogonal matrix |

The reliance on distractors for IPO also highlights a limitation of IRM: if a model can achieve a solution with zero training error, then any such solution is acceptable by IRM regardless of its generalization ability – a common scenario with overparametrized deep neural nets (Jin et al., 2020).

## H.1 NEAR-OPTIMALITY OF PSM AGGREGATION

**Conjecture 1.** *Assuming zero state aggregation error with policy similarity metric (PSM), the policy matrix $K$ learned using gradient descent is independent of the distractors.*

*Proof.* For LQR domains $x, y$, an observation pair $(o_t{}^x, o_t{}^y)$ has zero PSM iff the underlying state $s_t$ is same for both the observations in the pair. This is true, as (a) both domains has the same transition

dynamics, as specified by Equation H.1, and (b) the optimal policy is deterministic and is completely determined by the current state $s_t$ at any time $t$.

Assume $o_t{}^x = \begin{bmatrix} 0.1\ W_c \\ W_{d^x} \end{bmatrix} s_t$ and $o_t{}^y = \begin{bmatrix} 0.1\ W_c \\ W_{d^y} \end{bmatrix} s_t$ for distractor semi-orthogonal matrices $W_{d^x}$ and $W_{d^y}$, respectively. Furthermore, the representation is given by $K_1(o_t{}^x)$ and $K_1(o_t{}^y)$ respectively. Assume that $K_1 = [K_s \quad K_d]$ where $K_s \in \mathbb{R}^{h \times n_s}$ and $K_d \in \mathbb{R}^{h \times n_d}$ and $K_1 \in \mathbb{R}^{h \times (n_s + n_d)}$.

Zero state-aggregation error with squared loss implies that for pair $(o_t{}^x, o_t{}^y)$ corresponding to $s_t$,

$$K_1(o_t{}^x - o_t{}^y) = K_1 \begin{bmatrix} 0 \\ W_{d^x} - W_{d^y} \end{bmatrix} s_t = 0 \implies K_d(W_{d^x} - W_{d^y})s_t = 0 \qquad \text{(H.2)}$$

As Equation H.2 holds for all states visited by the optimal policy in an infinite horizon LQR, it follows that $K_d(W_{d^x} - W_{d^y}) = 0$.

Furthermore, it is well-known that gradient descent tends to find low-rank solutions due to implicit regularization (Arora et al., 2019; Gunasekar et al., 2017), *e.g.,* with small enough step sizes and initialization close enough to the origin, gradient descent on matrix factorization converges to the minimum nuclear norm solution for 2 layer linear networks (Gunasekar et al., 2017). Based on this, we conjecture that $K_d = 0$ which we found to be true in practice. □

# I   DISTRACTING CONTROL SUITE

We use the same setup as Kostrikov et al. (2020); Stone et al. (2021) for implementation details and training protocol. For completeness, we describe the details below.

**Dynamic Background Distractions**. In Distracting Control Suite (Stone et al., 2021), random backgrounds are projected from scenes of the DAVIS 2017 dataset (Pont-Tuset et al., 2017) onto the skybox of the scene. To make these backgrounds visible for all tasks and views, the floor grid is semi-transparent with transparency coefficient 0.3. We take the first 2 videos in the DAVIS 2017 training set and randomly sample a scene and a frame from those at the start of every episode. In the dynamic setting, the video plays forwards or backwards until the last or first frame is reached at which point the video is played backwards. This way, the background motion is always smooth and without "cuts".

**Soft Actor-Critic**. Soft Actor-Critic (SAC) (Haarnoja et al., 2018) learns a state-action value function $Q_\theta$, a stochastic policy $\pi_\theta$ and a temperature $\alpha$ to find an optimal policy for an MDP $(\mathcal{S}, \mathcal{A}, p, r, \gamma)$ by optimizing a $\gamma$-discounted maximum-entropy objective. $\theta$ is used generically to denote the parameters updated through training in each part of the model. The actor policy $\pi_\theta(a_t|s_t)$ is a parametric $\tanh$-Gaussian that given $s_t$ samples $a_t = \tanh(\mu_\theta(s_t) + \sigma_\theta(s_t)\epsilon)$, where $\epsilon \sim \mathcal{N}(0, 1)$ and $\mu_\theta$ and $\sigma_\theta$ are parametric mean and standard deviation.

The policy evaluation step learns the critic $Q_\theta(s_t, a_t)$ network by optimizing a single-step of the soft Bellman residual

$$J_Q(\mathcal{D}) = E_{\substack{(s_t, a_t, s'_t) \sim \mathcal{D} \\ a'_t \sim \pi(\cdot|s'_t)}}[(Q_\theta(s_t, a_t) - y_t)^2]$$
$$y_t = r(s_t, a_t) + \gamma[Q_{\theta'}(s'_t, a'_t) - \alpha \log \pi_\theta(a'_t|s'_t)],$$

where $\mathcal{D}$ is a replay buffer of transitions, $\theta'$ is an exponential moving average of the weights. SAC uses clipped double-Q learning, which we omit for simplicity but employ in practice.

The policy improvement step then fits the actor policy $\pi_\theta(a_t|s_t)$ network by optimizing the objective

$$J_\pi(\mathcal{D}) = E_{s_t \sim \mathcal{D}}[D_{\text{KL}}(\pi_\theta(\cdot|s_t) \| \exp\{\frac{1}{\alpha}Q_\theta(s_t, \cdot)\})].$$

Finally, the temperature $\alpha$ is learned with the loss

$$J_\alpha(\mathcal{D}) = E_{\substack{s_t \sim \mathcal{D} \\ a_t \sim \pi_\theta(\cdot|s_t)}}[-\alpha \log \pi_\theta(a_t|s_t) - \alpha\bar{\mathcal{H}}],$$

where $\bar{\mathcal{H}} \in \mathbb{R}$ is the target entropy hyper-parameter that the policy tries to match, which in practice is usually set to $\bar{\mathcal{H}} = -|\mathcal{A}|$.

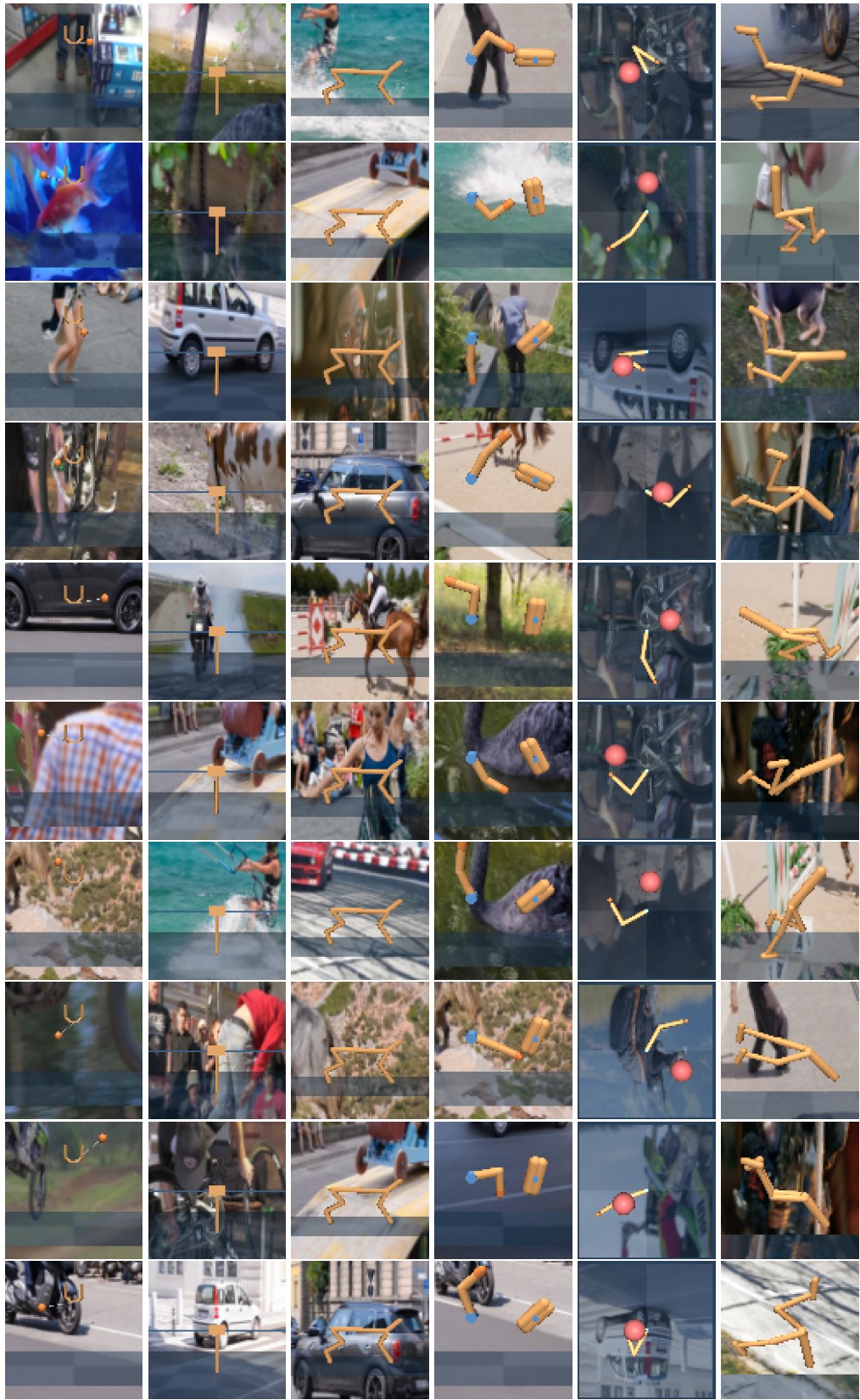

Figure I.1: **DCS Test Environments**: Snapshots of test environments used for evaluating generalization on Distracting Control Suite. Random backgrounds are projected from scenes of the first 30 videos of DAVIS 2017 validation dataset.

## I.1 ACTOR AND CRITIC NETWORKS

Following Kostrikov et al. (2020), we use clipped double Q-learning for the critic, where each $Q$-function is parametrized as a 3-layer MLP with `ReLU` activations after each layer except of the last. The actor is also a 3-layer MLP with `ReLU`s that outputs mean and covariance for the diagonal Gaussian that represents the policy. The hidden dimension is set to 1024 for both the critic and actor.

## I.2 ENCODER NETWORK

We employ the encoder architecture from Kostrikov et al. (2020). This encoder consists of four convolution layers with $3 \times 3$ kernels and 32 channels. The `ReLU` activation is applied after each convolutional layer. We use stride to 1 everywhere, except of the first convolutional layer, which has stride 2. The output of the convnet is feed into a single fully-connected layer normalized by `LayerNorm`. Finally, we apply `tanh` nonlinearity to the 50 dimensional output of the fully-connected layer. We initialize the weight matrix of fully-connected and convolutional layers with the orthogonal initialization and set the bias to be zero. The actor and critic networks both have separate encoders, although we share the weights of the conv layers between them. Furthermore, only the critic optimizer is allowed to update these weights (*i.e.,*we stop the gradients from the actor before they propagate to the shared convolutional layers).

## I.3 CONTRASTIVE METRIC EMBEDDING LOSS

For all our experiments, we use a single ReLU layer with $k = 256$ units for the non-linear projection to obtain the embedding $z_\theta$ (Figure A.1). We compute the embedding using the penultimate layer in the actor network. For picking the hyperparameters, we used 3 temperatures $[0.1, 0.01, 1.0]$ and 3 auxiliary $\mathscr{L}_{\mathrm{CME}}$ loss coefficients $[1, 3, 10]$ using "Ball In Cup Catch" as the validation environment. All other hyperparameters are the same as prior work (see Table I.2).

We approximate optimal policies with the policies obtained after training a DrQ agent for 500K environment steps. Since a given action sequence from this approximate policy has the same performance across different training environments, we compute the PSM across training environments, via dynamic programming (see Section J.1 for pseudo-code), using such action sequences.

**Total Loss.** The total loss is given by $\mathscr{L}_{\mathrm{RL}} + \alpha \mathscr{L}_{\mathrm{CME}}$ where $\mathscr{L}_{\mathrm{RL}}$ is the reinforcement learning loss which combines $J_\pi(\mathcal{D})$, $J_\pi(\mathcal{D})$, and $J_\alpha(\mathcal{D})$), while $\mathscr{L}_{\mathrm{CME}}$ is the auxiliary loss for learning PSEs with the coefficient $\alpha$.

## I.4 TRAINING AND EVALUATION SETUP

For evaluation, we use the first 30 videos from the DAVIS 2017 validation dataset (see Figure I.1). Each checkpoint is evaluated by computing the average episode return over 100 episodes from the unseen environments. All experiments are performed with five random seeds per task used to compute means and standard deviations/errors of their evaluations. We use $K = 2, M = 2$ as prescribed by Kostrikov et al. (2020) for DrQ. Following Kostrikov et al. (2020) and Stone et al. (2021), we use a different action repeat hyper-parameter for each task, which we summarize in Table I.3. We construct an observational input as a 3-stack of consecutive frames (Kostrikov et al., 2020), where each frame is an RGB rendering of size $84 \times 84$ from the 0th camera. We then divide each pixel by 255 to scale it down to $[0, 1]$ range. For data augmentation, we maintain temporal consistency by using the same crop augmentation across consecutive frames.

Table I.1: Optimal hyperparameters for PSE auxiliary loss for reporting results in Table 3.

| Hyperparameter | Setting |
|---|---|
| Contrastive temperature ($1/\lambda$) | 0.1 |
| Auxiliary loss coefficient ($\alpha$) | 1.0 |
| $\Gamma$-scale parameter ($\beta$) | 0.1 |
| Batch Size ($\mathscr{L}_{\mathrm{CME}}$) | 128 |
| $|\tau_{\mathcal{X}}^*|$ | 1000 // Action Repeat |

Table I.2: Hyper-parameters taken from Kostrikov et al. (2020) in the Distracting Control Suite experiments.

| Parameter | Setting |
|---|---|
| Replay buffer capacity | $100,000$ |
| Seed steps | $1,000$ |
| Batch size (DrQ) | $512$ |
| Discount $\gamma$ | $0.99$ |
| Optimizer | Adam |
| Learning rate | $10^{-3}$ |
| Critic target update frequency | $2$ |
| Critic Q-function soft-update rate $\tau$ | $0.01$ |
| Actor update frequency | $2$ |
| Actor log stddev bounds | $[-10, 2]$ |
| Init temperature | $0.1$ |

Table I.3: The action repeat hyper-parameter used for each task in the Distracting Control Suite benchmark.

| Task name | Action repeat |
|---|---|
| Cartpole Swingup | 8 |
| Reacher Easy | 4 |
| Cheetah Run | 4 |
| Finger Spin | 2 |
| Ball In Cup Catch | 4 |
| Walker Walk | 2 |

## I.5 GENERALIZATION CURVES

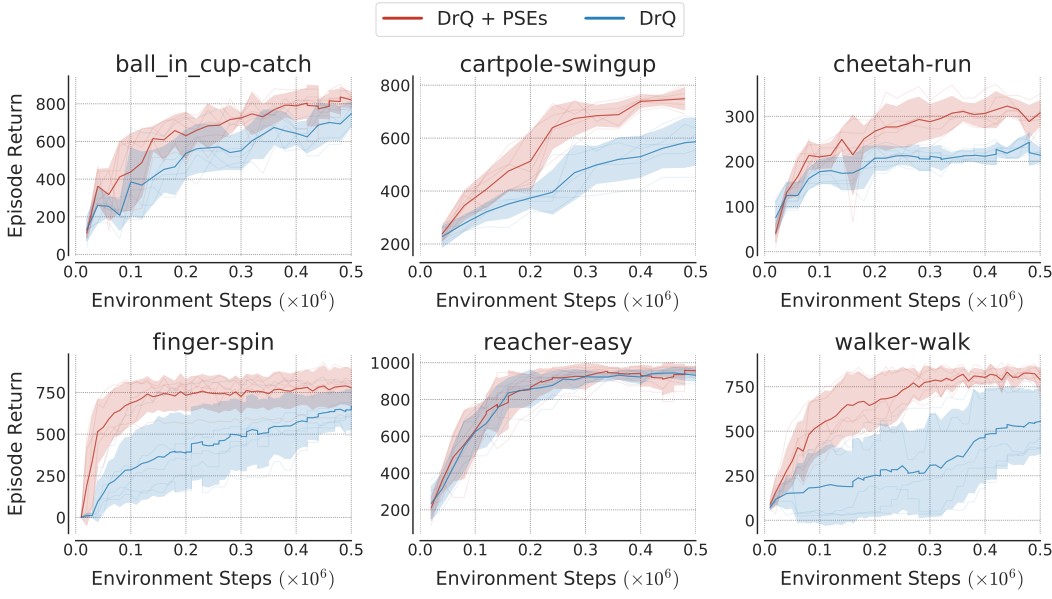

Figure I.2: **Random Initialization**. Generalization performance on unseen environments over the course of training. The agent is initialized randomly. DrQ augmented with PSEs outperforms DrQ. We plot the average episode return across 5 seeds and the shaded region shows the standard deviation. Each checkpoint is evaluated using 100 episodes with unseen backgrounds.

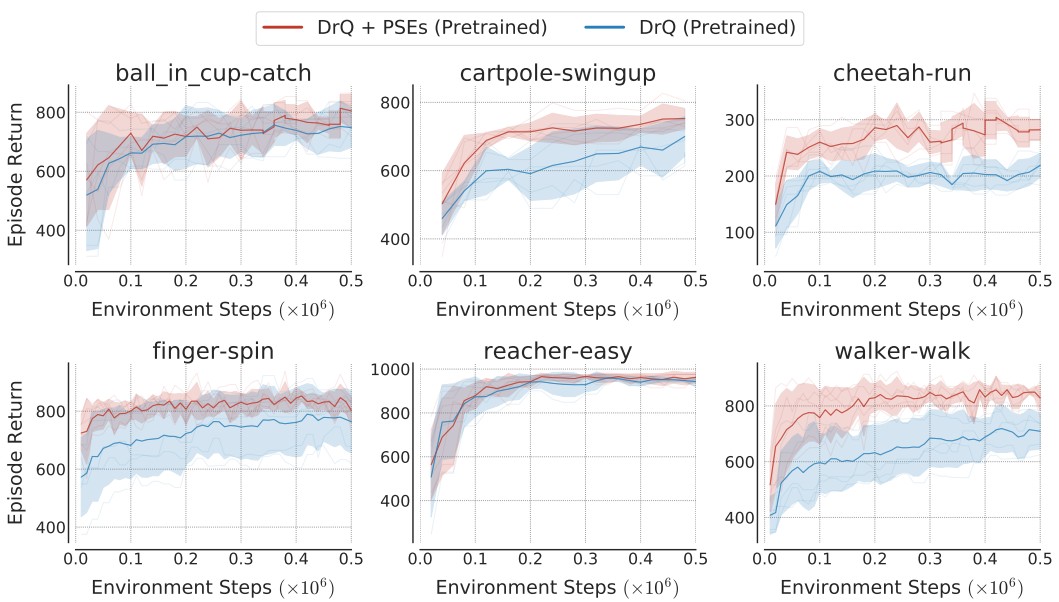

Figure I.3: **Pretrained Initialization**. Generalization performance on unseen environments over the course of training. The agent is initialized using a pretrained DrQ agent. DrQ augmented with PSEs outperforms DrQ on most of the environments. We plot the average return across 5 seeds and the shaded region shows the standard deviation. Each checkpoint is evaluated using 100 episodes with unseen backgrounds.

## J  PSEUDO CODE

### J.1  DYNAMIC PROGRAMMING FOR COMPUTING PSM

```python
def metric_fixed_point(cost_matrix, gamma=0.99, eps=1e-7):
  """DP for calculating PSM in environments with deterministic dynamics.

  Args:
    cost_matrix: DIST matrix where entries at index (i, j) is DIST(x_i,
    y_j)
    gamma: Metric discount factor.
    eps: Threshold for stopping the fixed point iteration.
  """
  d = np.zeros_like(cost_matrix)
  def operator(d_cur):
    d_new = 1 * cost_matrix
    discounted_d_cur = gamma * d_cur
    d_new[:-1, :-1] += discounted_d_cur[1:, 1:]
    d_new[:-1, -1] += discounted_d_cur[1:, -1]
    d_new[-1, :-1] += discounted_d_cur[-1, 1:]
    return d_new

  while True:
    d_new = operator(d)
    if np.sum(np.abs(d - d_new)) < eps:
      break
    else:
      d = d_new[:]
  return d
```

### J.2  CONTRASTIVE LOSS

```python
def contrastive_loss(similarity_matrix,
                     metric_values,
                     temperature,
```

```
4                       beta=1.0):
5     """Contrative Loss with embedding similarity ."""
6     metric_shape = tf.shape(metric_values)
7     ## z_\theta(X): embedding_1 = nn_model.representation(X)
8     ## z_\theta(Y): embedding_2 = nn_model.representation(Y)
9     ## similarity_matrix = cosine_similarity(embedding_1, embedding_2
10    ## metric_values = PSM(X, Y)
11    similarity_matrix /= temperature
12    neg_logits1 = similarity_matrix
13
14    col_indices = tf.cast(tf.argmin(metric_values, axis=1), dtype=tf.int32)
15    pos_indices1 = tf.stack(
16        (tf.range(metric_shape[0], dtype=tf.int32), col_indices), axis=1)
17    pos_logits1 = tf.gather_nd(similarity_matrix, pos_indices1)
18
19    metric_values /= beta
20    similarity_measure = tf.exp(-metric_values)
21    pos_weights1 = -tf.gather_nd(metric_values, pos_indices1)
22    pos_logits1 += pos_weights1
23    negative_weights = tf.math.log((1.0 - similarity_measure) + 1e-8)
24    neg_logits1 += tf.tensor_scatter_nd_update(
25        negative_weights, pos_indices1, pos_weights1)
26
27    neg_logits1 = tf.math.reduce_logsumexp(neg_logits1, axis=1)
28    return tf.reduce_mean(neg_logits1 - pos_logits1) # Equation 4
```

a

