# OpenReview forum: "Contrastive Behavioral Similarity Embeddings for Generalization in Reinforcement Learning"
_ICLR.cc/2021/Conference — ICLR 2021 Spotlight_

### Official Review · AnonReviewer1 · 2020-10-27
**Theoretically well-motivated approach and promising empirical results**

**Rating:** 6
**Confidence:** 3

**Review:**

Motivated by the issue of RL policy generalization, this paper explores improving generalization via a learned contrastive representation to embed states in, rather than through data augmentation or regularization alone. They demonstrate that their approach of policy similarity embeddings (PSEs) leads to improved generalization on several benchmarks, building on the method of policy bisimulation.

Strengths:
+ I think this approach is theoretically well-motivated, and the authors also give good intuition, especially in light of the latest work on self-supervised learning.
+ Experimentally, I thought the controls were well-chosen and illustrated the utility of their method on each task they considered (Figures 4 and Tables 1 and 3).

Weaknesses:
-	I would be curious to see how this approach works on more scaled up tasks with larger action spaces (Section 6.2 is in that direction, for instance), but the fact that it outperforms data augmentation techniques (including bisimulation transfer) in these simpler domains is promising.
-	In the jumping task, no intuition is given as to why PSE underperforms in the “narrow grid” (Table 1, first row) compared to bisimulation transfer, and it would be helpful to explain the failure cases of their method in more detail generally.

As it stands, I think the ideas of this paper are interesting and novel, but I would like to see tasks with larger action spaces or more naturalistic inputs than the ones considered here, and in the cases where the method does not perform well, documentation and potential intuition for why that might be. Therefore, I recommend a weak accept.

---

> ### Author Response · Authors · 2020-11-16
> **Author Response**
>
> We thank the reviewer for constructive feedback. To address their concerns, we have documented the reasons behind the reported performance of PSEs on “narrow” grid near the end of **Section 5.1**. Our responses follow:
>
> > **In the jumping task, no intuition is given as to why PSE underperforms in the “narrow grid” compared to bisimulation transfer, and it would be helpful to explain the failure cases of their method in more detail generally.**
>
> PSEs perform comparably to regularization on the “narrow” grid, which can be seen from the overlap of standard deviation in Table 1. In general, all learning-based methods are ineffective on the “narrow” grid without data augmentation. That said, PSEs do work quite well when combined with RandConv. However, even with data augmentation, generalization in “narrow” grid happens only around the training tasks, exhibiting the challenge this grid poses for learning-based methods. We believe this is due to the poor extrapolation ability of neural networks (e.g., Haley & Soloway, 1992; Xu et al., 2020), which is more perceptible without prior inductive bias from data augmentation.
>
> We emphasize that **bisimulation transfer is not a learning method and is impractical** for generalization as it requires oracle access to test rewards and dynamics. Moreover, we have added a comparison with bisimulation representations (Zhang et al, 2020) with RandConv in Table 1, showing it degrades performance by 30-50% compared to PSEs.
>
> > **How this approach works on more scaled up tasks with larger action spaces (Section 6.2 is in that direction, for instance)?**
>
> Distracting DM Control (Section 6.2) uses a continuous action space which is potentially infinite, so we are unsure what the reviewer means by larger action space. Similarly, the LQR benchmark (Section 6.1) uses 20 dimensional action space, which is as large as the action sets from common benchmarks such as Atari.

---

### Official Review · AnonReviewer2 · 2020-10-28

**Rating:** 6
**Confidence:** 3

**Review:**

This paper defines a new metric for calculating similarities between states. The policy similarity metric is basically based on the $\pi$-bisimulation metrics, but the l1 distance term of rewards is modified to the distance between states transitioned by the grounded policy. The experiments performed with several benchmarks, and the gain seems to be consistent.

[Quality]

The paper is clearly written overall and well arranged.

[Originality & Significance]

The originality and significance are sufficient for acceptance, as the authors devised a novel metric.

[Strengths]
+ Even though I didn’t thoroughly check all of the derivations, the metric seems to be more precise than $\pi$-bisimulation, similarity between states originated from different environments.
+ This approach can be easily applied to various RL models.

[Weaknesses]
- In Table 1, experiments with RandConv only contain RandConv and RandConv+PSEs. It should be better to compare to other methods, at least Bisimulation, to show how much the PSE model performs well. Not only Table 1, I feel overall comparison results are not sufficient.
- The Distracting Control Suite dataset, which is the core dataset in the experiments, is not currently available, so verifying the results with this dataset is difficult.

[Questions or Additional Comments]
1. Figure 2 is not self-explanatory and hard to understand. I think adding axes or an additional explanation in the caption is required.
2. Please elaborate on the reason for using the Distracting DM Control Suite dataset rather than other datasets.
3. According to Tables G.3 and G.4, hyperparameters are quite different in each experiment setting. Does the proposed method’s performance highly depend on hyperparameter settings?

---

> ### Author Response · Authors · 2020-11-16
> **Author Response**
>
> We thank the reviewer for their constructive feedback. As requested, we added a comparison with bisimulation, when using  Randconv in **Table 1** (2nd row) and clarify about robustness of PSEs to hyperparameter tuning and use of Distracting DM control. Our responses follow:
>
> > **In Table 1 .. It should be better to compare to other methods, at least Bisimulation, to show how much the PSE model performs well.**
>
> We have added the comparison with **RandConv + Bisimulation** representations ([Zhang et al., 2020](https://arxiv.org/abs/2006.10742)) in Table 1  which shows that bisimulation **degrades performance by 30-50%** compared to PSEs. We have also highlighted the comparison in Table 2, where we use bisimulation with CMEs.
>
> > **Please elaborate on the reason for using the Distracting DM Control Suite dataset rather than other datasets.**
>
> *  Distracting DM Control (DCS) tests whether agents can ignore high-dimensional visual distractors irrelevant to the RL task. This benchmark uses continuous control actions without having access to optimal policies, which shows the scalability and applicability of PSEs beyond discrete actions.
>
>
> *  DM Control has been used for evaluating generalization, e.g., [RAD](https://arxiv.org/abs/2004.14990) (Laskin et al., 2020), [CURL](https://arxiv.org/abs/2004.04136) (Srinivas et al., 2020), [DrQ](https://arxiv.org/abs/2004.13649) (Kostrikov et al., 2020) etc. However, these methods are not sufficient to solve DCS — for example, DrQ nearly solves DM control but is suboptimal on DCS as shown in Table 3. Furthermore, background distractions have been proposed previously ([Zhang et al., 2018](https://arxiv.org/abs/1811.06032)) as well as utilized by concurrent ICLR submissions ([Anonymous 2020a](https://openreview.net/forum?id=-2FCwDKRREu), [Anonymous 2020a](https://openreview.net/forum?id=aa0705s2Qc)).
>
> > **The Distracting Control Suite dataset, is not currently available, so verifying the results with this dataset is difficult.**
>
> We mention all the details of how the Distracting DM Control environments are created as well as how the experiments were conducted in Appendix I.  Additionally, we will open source our code when DCS is publicly released.
>
> > **Figure 2 is not self-explanatory and hard to understand.**
>
> We updated the caption as well as added labels to improve this figure.
>
> > **According to Tables G.3 and G.4, hyperparameters are quite different in each experiment setting. Does the proposed method’s performance highly depend on hyperparameter settings?**
>
> We found PSEs to be quite robust to hyperparameters.
>
> Evaluating PSEs with the jumping task hyperparameters from Table G.3 instead of the ones in Table G.4 leads to a small drop (-4%) in performance on the jumping task with colors (Section 5.4). Nevertheless, PSEs still outperform other methods in Section 5.4. Additionally, we did not tune hyperparameters for every task, for example, we simply used parameters tuned on the wide grid for the narrow grid and random grid in Table 1. Similarly, we only tuned 3 choices each for temperature and auxiliary loss coefficient on the DM control experiments, as mentioned in Appendix I.3.
>
> We added more details about hyperparameter tuning in Appendix G.

---

### Official Review · AnonReviewer3 · 2020-10-28
**Well-motivated representation learning approach for RL generalization**

**Rating:** 7
**Confidence:** 3

**Review:**

Summary:

This work proposes a policy similarity metric (PSM) which differs slightly from the \pi-bisimulation metric: while the \pi-bisimulation metric defines the distance between two states in terms of the difference between the expected rewards obtained by following policy \pi, the proposed policy similarity metric measures distance based on the similarity between the optimal policies at each state. The policy similarity metric can then be used to learn policy similarity embeddings (PSEs) via contrastive representation learning. These embeddings should in principle improve generalization in RL.



Strengths:

- The proposed policy similarity metric is well motivated for generalization in RL, which is an important problem. I also appreciate the theoretical upper bound on the suboptimality of transferring policies to a new environment.

- The experiments are thorough and explore very diverse tasks. They highlight how PSEs can be used in imitation learning, LQR under spurious correlations, and RL settings; and can be combined with existing data augmentation techniques, including RandConv and DrQ.

- The ablation studies and visualizations demonstrate, both quantitatively and qualitatively, the individual importance of the policy similarity metric and the contrastive metric embeddings.

- The paper is very clearly written and well structured.



Weaknesses:

- The main weakness of the proposed policy similarity metric is that its computation requires optimal trajectories or approximately optimal trajectories (obtained, for example, through RL) from at least two training MDPs.

- Building upon the previous point, I am curious about the utility of PSEs learned from increasingly suboptimal policies. I think including such a study would greatly help understand the sensitivity of learning PSEs to quality of the policies.

- In settings like the jumping task from pixels, because of the reliance of the PSM on positive examples, data augmentation is required for good performance.



Recommendation:

I am recommending to accept this paper. I think the work is well motivated, approaches an important problem, and conducts very thorough experiments. I would still like to understand how useful PSEs are when learned from increasingly suboptimal policies.



Questions:

- For the Distracting Control Suite tasks, are the “DrQ + PSEs” policies trained with both DrQ and PSEs or only PSEs? Specifically, I’m trying to understand if the gains are only from PSEs as opposed to the combination of PSEs and data augmentation.



Updates:

Thank you for the clarifications, new study, and theoretical justification. The new results demonstrate the utility of PSEs even when policies are not exactly optimal. My concerns were thoroughly addressed by the authors’ response, and I’ve updated my score accordingly.

---

> ### Author Response · Authors · 2020-11-16
> **Author Response**
>
> We thank the reviewer for their constructive feedback. To address their concerns, we have added a study about the utility of PSEs learned from suboptimal policies of varying degree **(Section 5.3)** as well as a theoretical result bounding PSM approximation error **(Proposition D.3)**. We also clarified the orthogonality of PSEs to data augmentation. Our responses follow:
>
> >  **Utility of PSEs learned from increasingly suboptimal policies .. including a study would greatly help understand the sensitivity of learning PSEs to the quality of the policies.**
>
> We have added this study (Figure 4) on jumping task where we use increasingly suboptimal policies varying from the optimal policy to the uniform random policy. This study confirms that the utility of PSEs for generalization is robust to suboptimality.  One reason for this robustness is that PSEs are likely to align state representations with similar long-term greedy optimal actions, resulting in good performance even with suboptimal policies that preserve these greedy actions.
>
> In addition, we theoretically show (Proposition D.3) that the error in approximate PSM ($\hat{d}$) computed using a suboptimal policy is bounded by the policy’s suboptimality (based on a generalized notion of PSM). Thus, for policies with decreasing suboptimality, the PSM approximation becomes more accurate, resulting in improved PSEs.
> $$|d^*(x, y) - \hat{d}(x, y)|\  <  \underbrace{d\big((x, \pi^*), (x, \hat{\pi})\big)}_{\text{Long-term suboptimality difference from $x$}} + d\big((y, \pi^*), (y, \hat{\pi})\big)$$
>
> > **PSEs require data augmentation is required for good performance**
>
> PSEs don't require data augmentation but are complementary and can be combined with data augmentation when convenient. Moreover, data augmentation can degrade performance too, as shown by **Section 5.4**, where PSEs without any augmentation perform the best.
>
> >  **In settings like the jumping task from pixels, because of the reliance of the PSM on positive examples ...**
>
> PSEs do not rely on positive examples. The small number of positive examples on the jumping task results in a highly unbalanced classification problem with low amounts of data (recall we are doing imitation learning), making it challenging without additional inductive biases.  Using data augmentation for good performance does not indicate any limitation of PSEs but rather shows the difficulty of solving jumping task without prior inductive biases. We have clarified this in the revised paper.
>
> > **For the Distracting Control Suite, .. are gains only from PSEs as opposed to the combination of PSEs and data augmentation**
>
> The policies are trained jointly with PSEs and DrQ (= SAC + data augmentation). DrQ heavily utilizes data augmentation and we simply apply PSEs on using the same augmentation as DrQ (Figure A.1). The benefits from PSEs over DrQ stem from the alignment of state representations across different environments (Equation 4).

---

### Official Review · AnonReviewer4 · 2020-10-28
**Official Blind Review #4**

**Rating:** 7
**Confidence:** 3

**Review:**

The authors propose an interesting direction toward generalization in reinforcement learning. The authors propose a policy similarity metric (PSM) for measuring behavioral similarity between states. States are considered similar if the optimal policies in those states (as well as future states) are similar. The authors present a contrastive representation procedure to embed state similarity metrics, which are instantiated with PSM to yield policy similarity embeddings (PSEs).

I like the idea of looking at the similarity between policies by examining the difference in local and long-term behavior. Furthermore, I like that the contrastive metric embedding approach is relatively simple and that PSEs can be learned alongside RL using an auxiliary objective.

Questions and comments:
1. What informed the train test split in the problem setup on page 4?
2. I would like to see how this approach performs on environments specifically designed to test generalization, such as the ProcGen [1] and MineRL [2] environments.

[1] Cobbe, et al. Leveraging procedural generation to benchmark reinforcement learning. arXiv:1912.01588, 2019.
[2] Guss, et al. The MineRL competition on sample efficient reinforcement learning using human priors. arXiv:1904.10079, 2019.

---

> ### Author Response · Authors · 2020-11-16
> **Author Response**
>
> We thank the reviewer for their constructive feedback. We have included the rationale for the test train split on the jumping task in the revised paper. Our responses follow:
>
> > **What informed the train test split in the problem setup on page 4?**
>
> For the jumping task, we wanted to stress test generalization using a few variations of the underlying factors (e.g., 6 obstacle positions and 3 floor heights for the “wide“/“narrow“ grid). The different grids capture different types of generalization:
>
> * ''Wide'' grid tests generalization in the form of **“interpolation”**: Given training positions and heights, can agents generalize to unseen positions and height combinations between these combinations?
>
> * ''Narrow'' grid tests out-of-distribution generalization in the form of **“extrapolation”**: Can agents generalize to unseen combinations outside of the range of training combinations?
>
> * Random grid instances test for **standard generalization** akin to supervised learning where training and test splits are sampled i.i.d. from the same distribution.
>
> >  **I would like to see how this approach performs on environments specifically designed to test generalization, such as the ProcGen [1] and MineRL [2] environments.**
>
> All environments in the paper are specifically designed to test generalization. Furthermore, each environment highlights a specific aspect of PSEs for improving generalization.
>
> * The **jumping task** (des Combes et al., 2018) captures, using well-defined factors of variations, whether agents can learn the correct invariances required for generalization, directly from image inputs?
>
> * **Distracting DM Control** (Anonymous 2020, Zhang et al. 2018) tests whether agents can ignore high-dimensional visual distractors irrelevant to the RL task. This benchmark allows us to show that PSEs can scale to RL settings, with continuous action spaces, in which we only have access to good policies.
>
> * The **LQR task** (Song et al., 2020) requires agents to avoid overfitting to spurious correlations which hurt generalization. We used LQR to show that the proposed approach, as opposed to data augmentation, is also applicable to non-image based RL domains.
>
> Finally, PSEs outperform strong data augmentation methods, which are current state-of-the-art on benchmarks such as ProcGen.

---

### Author Response · Authors · 2020-11-16
**Updated manuscript with new experiments and clarifications**

We thank the reviewers for their valuable feedback! All the reviewers found the paper to be clearly written and that it proposes a novel and theoretically well-motivated approach for generalization with promising empirical results in diverse domains.  Based on reviewers’ comments, we have added some new experiments as well as additional information (both shown in blue) and clarifications (highlighted in red) in the revised paper. We summarize the main changes below:

* *Robustness of PSEs to suboptimality*: We added a study which demonstrates the utility of PSEs learned from suboptimal policies of varying degree (Section 5.3) as well as a theoretical result bounding PSM approximation error (Proposition D.3)  *[Reviewer 1]*.


*   *Additional bisimulation comparisons*: We added a comparison (Table 1) with bisimulation when using RandConv, which shows that bisimulation degrades performance by 30-50% compared to PSEs *[Reviewer 2]*.  We also clarified the comparison with bisimulation in the ablations (Section 5.2) as well as the impracticality of bisimulation transfer  *[Reviewer 2, 4]*.


*   *Intuition for PSEs on narrow grid*: We explain our hypothesis behind the reported performance of PSEs and other learning-based methods on “narrow” grid (Section 5.1) *[Reviewer 4]*.


*  *Evaluation and benchmark choices*: We included the rationale for the test train split and different grids on the jumping task *[Reviewer 1]* as well as highlight the purpose of the benchmarks in the paper *[Reviewer 1, 3]*.


*  *Minor fix*: We corrected a reporting error on performance of bisimulation + l2-embeddings in Table 5.2 which doesn’t change any relative comparisons -- PSEs still outperforms bisimulation by more than 40%.

We would appreciate it if the reviewers can confirm that their concerns had been addressed and, if so, reconsider their assessment.

---

### Decision · Program_Chairs · 2021-01-07
**Final Decision**

**Decision:**

Accept (Spotlight)

**Comment:**

The paper describes a new technique to generalize across different environments.  More precisely a new state similarity metric is defined with a contrastive learning embedding technique.  Unlike previous works that extend supervised learning techniques such as data augmentation and regularization to RL, the proposed approach takes into account the sequential nature of RL.    The reviewers unanimously praised the work in terms of theory, algorithm and empirical evaluation.  This is a novel and technically deep contribution that advances the state of the art for RL generalization.